# PeriodWave: Multi-Period Flow Matching for High-Fidelity Waveform Generation

**Sang-Hoon Lee**[1,2,*]                **Ha-Yeong Choi**[3,*]                **Seong-Whan Lee**[4,†]

[1]Department of Software and Computer Engineering, Ajou University, Suwon, Korea
[2]Department of Artificial Intelligence, Ajou University, Suwon, Korea
[3]Gen AI Lab, KT Corp., Seoul, Korea
[4]Department of Artificial Intelligence, Korea University, Seoul, Korea

## Abstract

Recently, universal waveform generation tasks have been investigated conditioned on various out-of-distribution scenarios. Although one-step GAN-based methods have shown their strength in fast waveform generation, they are vulnerable to train-inference mismatch scenarios such as two-stage text-to-speech. Meanwhile, diffusion-based models have shown their powerful generative performance in other domains; however, they stay out of the limelight due to slow inference speed in waveform generation tasks. Above all, there is no generator architecture that can explicitly disentangle the natural periodic features of high-resolution waveform signals. In this paper, we propose *PeriodWave*, a novel universal waveform generation model from Mel-spectrogram and neural audio codec. First, we introduce a period-aware flow matching estimator that effectively captures the periodic features of the waveform signal when estimating the vector fields. Additionally, we utilize a multi-period estimator that avoids overlaps to capture different periodic features of waveform signals. Although increasing the number of periods can improve the performance significantly, this requires more computational costs. To reduce this issue, we also propose a single period-conditional universal estimator that can feed-forward parallel by period-wise batch inference. Additionally, we first introduce FreeU to reduce the high-frequency noise for waveform generation. Furthermore, we demonstrate the effectiveness of the proposed method in neural audio codec decoding task, and present the streaming generation framework of non-autoregressive model for speech language models. The experimental results demonstrated that our model outperforms the previous models in reconstruction tasks from Mel-spectrogram and discrete token, and text-to-speech tasks. Source code is available at `https://github.com/sh-lee-prml/PeriodWave`.

## 1 Introduction

Deep generative models have achieved significant success in high-fidelity waveform generation. In general, the neural waveform generation model which is called *"Neural Vocoder"* transforms a low-resolution acoustic representation such as Mel-spectrogram or linguistic representations into a high-resolution waveform for regeneration learning (Tan et al., 2024). Conventional neural vocoders have been investigated for text-to-speech (Oord et al., 2016; Shen et al., 2018; Ren et al., 2019; Kim et al., 2020; Jiang et al., 2024) and voice conversion (Lee et al., 2021; Choi et al., 2021). Furthermore, recent universal waveform generation models called *"Universal Vocoder"* are getting more attention due to their various applicability in neural audio codec (Zeghidour et al., 2021; Défossez et al., 2023; Kumar et al., 2024; Ju et al., 2024; Zhang et al., 2024), audio generation (Kreuk et al., 2023; Roman et al., 2023; Yang et al., 2023c; Huang et al., 2023; Liu et al., 2023), and zero-shot voice cloning systems (Lee et al., 2022d; Huang et al., 2022c; Wang et al., 2023; Li et al., 2024; Le et al., 2024; Kim et al., 2024; Shen et al., 2024) where models can generate high-fidelity waveform from the

---

*Equal contribution
†Corresponding author

highly compressed representations beyond the traditional acoustic features, Mel-spectrogram. In addition, universal vocoder requires generalization in various out-of-distribution scenarios including unseen voice, instruments, and dynamic environments (Lee et al., 2023; Bak et al., 2023).

Previously, generative adversarial networks (GAN) models dominated the waveform generation tasks by introducing various discriminators that can capture the different characters of waveform signals. MelGAN (Kumar et al., 2019) used the multi-scale discriminator to capture different features from the different scales of waveform signal. HiFi-GAN (Kong et al., 2020) introduced the multi-period discriminator to capture the different periodic patterns of the waveform signal. UnivNet (Jang et al., 2021) utilized the multi-resolution spectrogram discriminator that can reflect the spectral features of waveform signal. BigVGAN (Lee et al., 2023) proposed the Snake activation function for the out-of-distribution modeling and scaled up the neural vocoder for universal waveform generation. Vocos (Siuzdak, 2024) significantly improved the efficiency of the neural vocoder without upsampling the time-axis representation. Although GAN-based models can generate the high-fidelity waveform signal fast, GAN models possess three major limitations: 1) they should utilize a lot of discriminators to improve the audio quality, which increases training time; 2) this also requires hyper-parameter tuning to balance multiple loss terms; 3) they are vulnerable to train-inference mismatch scenarios such as two-state models, which induces metallic sound or hissing noise.

Recently, the multi-band diffusion (MBD) model (Roman et al., 2023) sheds light on the effectiveness of the diffusion model for high-resolution waveform modeling. Although previous diffusion-based waveform models (Kong et al., 2021; Chen et al., 2021) existed, they could not model the high-frequency information so the generated waveform only contains low-frequency information. Additionally, they still require a lot of iterative steps to generate high-fidelity waveform signals. To reduce this issue, PriorGrad (Lee et al., 2022b) introduced a data-driven prior and FastDiff (Huang et al., 2022a) adopted an efficient structure and noise schedule predictor. However, they do not model the high-frequency information so these models only generate the low-frequency information well.

Above all, there is no generator architecture to reflect the natural periodic features of high-resolution waveform signals. In this paper, we propose PeriodWave, a novel waveform generation model that can reflect different implicit periodic representations. We also adopt the powerful generative model, flow matching that can estimate the vector fields directly using the optimal transport path for fast sampling. Additionally, we utilize a multi-period estimator by adopting the prime number to avoid overlaps. We observed that increasing the number of periods can improve the entire performance consistently. However, this also induces a slow inference speed. To simply reduce this limitation, we propose a period-conditional universal estimator that can feed-forward parallel by period-wise batch inference. For high-frequency information modeling, we investigate discrete wavelet transformation (DWT) (Lee et al., 2022c) and FreeU in waveform generation tasks.

PeriodWave achieves a better performance in objective and subjective metrics than other publicly available strong baselines on both speech and out-of-distribution samples. Specifically, the experimental results demonstrated that our methods can significantly improve the pitch-related metrics including pitch distance, periodicity, and V/UV F1 score with unprecedented performance. Furthermore, we only train the models for only three days while previous GAN models require over three weeks.

Furthermore, we demonstrate the effectiveness of proposed method in neural audio codec decoding tasks using state-of-the-art neural audio codec, Mimi (Défossez et al.). Based on the tokens from Mimi, PeriodWave significantly improve audio quality compared to the original decoder of Mimi. Moreover, we propose the streaming generation method using PeriodWave trained by parallel generation, and PeriodWave can successfully decode in a stream manner with a minimal degradation.

The main contributions of this study are as follows:

- We propose PeriodWave, a novel universal waveform generator that can reflect different implicit periodic information when estimating the vector fields.
- We thoroughly analyze the limitation of high-frequency modeling, and we address this limitation by DWT and FreeU approach for high-frequency noise reduction.
- PeriodWave outperformed the one-step GAN models in conventional two-stage TTS tasks. Through iterative refinement, PeriodWave can reduce the train-inference mismatch problem.
- Based on SOTA neural audio codec Mimi, we successfully demonstrate the effectiveness of PeriodWave in neural audio codec decoding task both in parallel and streaming generation.

## 2 RELATED WORKS

**Neural Vocoder** WaveNet (Oord et al., 2016) has successfully paved the way for high-quality neural waveform generation tasks. However, these auto-regressive (AR) models suffer from a slow inference speed. To address this limitation, teacher-student distillation-based inverse AR flow methods (Oord et al., 2018; Ping et al., 2019) have been investigated for parallel waveform generation. Flow-based models (Kim et al., 2019; Prenger et al., 2019; Lee et al., 2020) have also been utilized, which can be trained by simply maximizing the likelihood of the data using invertible transformation.

**GAN-based Neural Vocoder** MelGAN (Kumar et al., 2019) successfully incorporated generative adversarial networks (GAN) into the neural vocoder by introducing a multi-scale discriminator to reflect different features from the different scales of waveform signal and feature matching loss for stable training. Parallel WaveGAN (Yamamoto et al., 2020) introduces multi-resolution STFT losses that can improve the perceptual quality and robustness of adversarial training. GAN-TTS (Bińkowski et al., 2020) utilized an ensemble of random window discriminators that operate on random segments of waveform signal. GED (Gritsenko et al., 2020) proposed a spectral energy distance with unconditional GAN for stable and consistent training. HiFi-GAN (Kong et al., 2020) introduced a novel discriminator, a multi-period discriminator (MPD) that can capture different periodic features of waveform signal. UnivNet (Jang et al., 2021) employed adversarial feedback on the multi-resolution spectrogram to capture the spectral representations at different resolutions. BigVGAN (Lee et al., 2023) adopted periodic activation function and anti-aliased representation into the generator for generalization on out-of-distribution samples. Vocos (Siuzdak, 2024) proposed an efficient waveform generation framework using ConvNeXt blocks and iSTFT head without any temporal domain upsampling. Meanwhile, neural codec models (Zeghidour et al., 2021; Défossez et al., 2023; Kumar et al., 2024) and applications (Wang et al., 2023; Yang et al., 2023b) such as TTS and audio generation have been investigated together with the development of neural vocoder.

**Diffusion-based Neural Vocoder** DiffWave (Kong et al., 2021) and WaveGrad (Chen et al., 2021) introduced a Mel-conditional diffusion-based neural vocoder that can estimate the gradients of the data density. PriorGrad (Lee et al., 2022b) improves the efficiency of the conditional diffusion model by adopting a data-dependent prior distribution for diffusion models instead of a standard Gaussian distribution. FastDiff (Huang et al., 2022a) proposed a fast conditional diffusion model by adopting an efficient generator structure and noise schedule predictor. Multi-band Diffusion (Roman et al., 2023) incorporated multi-band waveform modeling into diffusion models and it significantly improved the performance by band-wise modeling because previous diffusion methods could not model high-frequency information, which only generated the low-frequency representations. This model also focused on raw waveform generation from discrete tokens of neural codec model for various audio generation applications including speech, music, and environmental sound.

## 3 PERIODWAVE

The flow matching model (Lipman et al., 2022; Tong et al., 2023) has emerged as an effective strategy for the swift and simulation-free training of continuous normalizing flows (CNFs), producing optimal transport (OT) trajectories that are readily incorporable. We are interested in the use of flow matching models for waveform generation to understand their capability to manage complex transformations across waveform distributions. Hence, we begin with the essential notation to analyze flow matching with optimal transport, followed by a detailed introduction to the proposed method.

### 3.1 PRELIMINARY: FLOW MATCHING WITH OPTIMAL TRANSPORT PATH

In the data space $\mathbb{R}^d$, consider an observation $x \in \mathbb{R}^d$ sampled from an unknown distribution $q(x)$. CNFs transform a simple prior $p_0$ into a target distribution $p_1 \approx q$ using a time-dependent vector field $v_t$. The flow $\phi_t$ is defined by the ordinary differential equation:

$$\frac{d}{dt}\phi_t(x) = v_t(\phi_t(x); \theta), \quad \phi_0(x) = x, \quad x \sim p_0, \tag{1}$$

The flow matching objective, as introduced by (Lipman et al., 2022), aims to match the vector field $v_t(x)$ to an ideal vector field $u_t(x)$ that would generate the desired probability path $p_t$. The flow matching training objective involves minimizing the loss function $L_{FM}(\theta)$, which is defined by regressing the model's vector field $v_\theta(t, x)$ to a target vector field $u_t(x)$ as follows:

$$\mathcal{L}_{FM}(\theta) = \mathbb{E}_{t \sim [0,1], x \sim p_t(x)} ||v_\theta(t, x) - u_t(x)||_2^2. \tag{2}$$

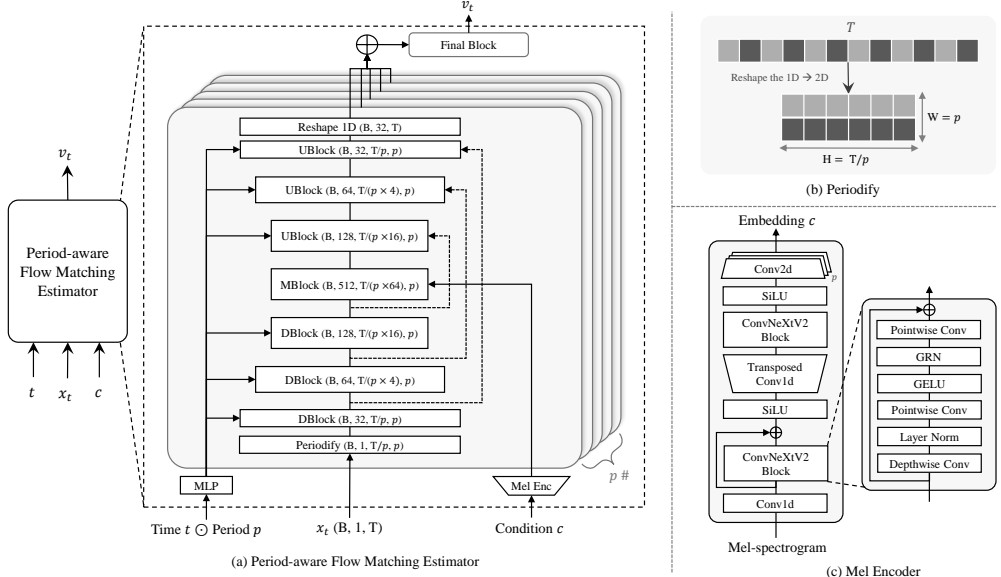

Figure 1: Waveform generation using conditional flow matching and ODE solver

Figure 2: Overall architecture of PeriodWave

Given the impracticality of accessing $u_t$ and $p_t$, conditional flow matching (CFM) is introduced:

$$\mathcal{L}_{CFM}(\theta) = \mathbb{E}_{t\sim[0,1],x\sim p_t(x|z)} ||v_\theta(t,x) - u_t(x|z)||_2^2. \tag{3}$$

Generalizing this with the noise condition $x_0 \sim N(0,1)$, the OT-CFM loss is:

$$L_{\text{OT-CFM}}(\theta) = \mathbb{E}_{t,q(x_1),p_0(x_0)}||u_t^{\text{OT}}(\phi_t^{\text{OT}}(x_0) \mid x_1) - v_t(\phi_t^{\text{OT}}(x_0) \mid \mu; \theta)||^2, \tag{4}$$

where $\phi_t^{\text{OT}}(x_0) = (1 - (1 - \sigma_{\min})t)x_0 + tx_1$ and $u_t^{\text{OT}}(\phi_t^{\text{OT}}(x_0) \mid x_1) = x_1 - (1 - \sigma_{\min})x_0$. This approach efficiently manages data transformation and enhances training speed and efficiency by integrating optimal transport paths. The detailed formulas are described in Appendix A.

## 3.2 PERIOD-AWARE FLOW MATCHING ESTIMATOR

In this work, we propose a period-aware flow matching estimator, which can reflect the different periodic features when estimating the vector field for high-quality waveform generation as illustrated in Figure 1. First, we utilize a time-conditional UNet-based structure for time-specific vector field estimation. Unlike previous UNet-based decoders, PeriodWave utilizes a mixture of reshaped input signals with different periods as illustrated in Figure 2. Similar to (Kong et al., 2020), we reshape the 1D data sampled from $p_t(x)$ of length $T$ into 2D data of height $T/p$ and width $p$. We will refer to this process as *Periodify*. Then, we condition the period embedding to indicate the specific period of each reshaped sample for period-aware feature extraction in a single estimator. We utilize different periods of [1,2,3,5,7] that avoid overlaps to capture different periodic features from the input signal. We utilize 2D convolution of down/upsampling layer and ResNet Blocks with a kernel size of 3 and dilation of 1, 2 for each UNet block. Specifically, we downsample each signal by [4,4,4] so the representation of the middle block has height $T/(p \times 64)$ and width $p$. After extracting the representation for each period, we reshape the 2D representation into the original shape of the 1D signal for each period path. We sum all representations from all period paths. The final block estimates the vector fields from a mixture of period representations.

For Mel-spectrogram conditional generation, we only add the conditional representation extracted from Mel-spectrogram to the middle layer representation of UNet for each period path. We utilize

ConvNeXt V2 based Mel encoder to extract the conditional information for efficient time-frequency modeling. Previously, Vocos (Siuzdak, 2024) also demonstrated that ConvNeXt-based time-frequency modeling shows effectiveness on the low resolution features. In this works, we utilize the improved ConvNeXt V2 (Woo et al., 2023) blocks for Mel encoder, and the output of this block is fed to the period-aware flow matching estimator. Because we utilize a hop size of 256, the Mel-spectrogram has a length of $T/256$. To align the conditional representation, we upsample it by 4× and downsample it by the different strides as periods of [1,2,3,5,7] to get a shape of $T/(p \times 64)$.

To boost the inference speed, we introduce two methods: 1) period-wise batch inference that can feed-forward parallel for multiple periods by a period-conditional universal estimator; 2) time-shared conditional representation extracted from Mel-spectrogram, which is utilized for every step.

## 3.3 FLOW MATCHING FOR WAVEFORM GENERATION

To the best of our knowledge, this is the first work to utilize flow matching for waveform generation. In this subsection, we describe the problems we encountered and how to reduce these issues. First, we found that the it is crucial to set the proper noise scale for $x_0$. In general, waveform signal is ranged with -1 to 1, so standard normal distribution $\mathcal{N}(0, 1)$ would be large for optimal path. This results in high-frequency information distortion, causing the generated sample to contain only low-frequency information. To reduce this issue, we scale down the $x_0$ by multiplying a small value $\alpha$. Although we successfully generate the waveform signal by small $\alpha$, we observed that the generated sample sometimes contains a small white noise. We simply solve it by additionally multiplying temperature $\tau$ on the $x_0$ as analyzed in Table 4. Furthermore, we adopt data-dependent prior (Lee et al., 2022b) to flow matching-based generative models. Specifically, we utilize an energy-based prior which can be simply extracted by averaging the Mel-spectrogram along the frequency axis. We set $\mathcal{N}(0, \Sigma)$ for the distribution of $p_0(x)$, and multiply $\Sigma$ by a small value of 0.5. All of them significantly improve the sample quality and boost the training speed.

## 3.4 HIGH-FREQUENCY INFORMATION MODELING FOR FLOW MATCHING

Similar to the findings demonstrated by (Roman et al., 2023), we also observed that flow matching-based waveform generation models could not provide the high-frequency information well. To address this limitation, we adopt three approaches including multi-band modeling and FreeU (Si et al., 2024)

**Multi-band Flow Matching with Discrete Wavelet Transform** Previously, MBD (Roman et al., 2023) demonstrated that diffusion-based models are vulnerable to high-frequency noise so they introduce the multi-band diffusion models by disentangling the frequency bands and introducing specialized denoisers for each band. Additionally, they proposed frequency equalizer (EQ) processor to reduce the white noise by regularizing the noise energy scale for each band.[1] Unlike MBD, we introduce a discrete wavelet Transform based multi-band modeling method which can disentangle the signal and reproduce the original signal without losing information[2]. PeriodWave-MB consists of multiple vector field estimators for each band [0-3, 3-6, 6-9, 9-12 kHz]. Additionally, we first generate a lower band, and then concatenate the generated lower bands to the $x_0$ to generate higher bands. We found that this significantly improve the quality even with small sampling steps. During training, we utilize a ground-truth discrete wavelet Transform components for a conditional information. Additionally, we also utilize a band-wise data-dependent prior by averaging Mel-spectrogram according to the frequency axis including overlapped frequency bands [0-61, 60-81, 80-93, 91-100 bins]. Moreover, we downsample each signal by [1,4,4] by replacing the first down/up-sampling with DWT/iDWT, and this also significantly reduce the computational cost by reducing time resolution.

**Flow Matching with FreeU** FreeU (Si et al., 2024) demonstrated that the features from the skip connection contain high-frequency information in UNet-based diffusion models, and this could ignore the backbone semantics during image generation. We revisited this issue in high-resolution

---

[1]We entirely acknowledged (Roman et al., 2023) proposed a novel pre-processing method and combined it with diffusion models well. However, we do not use any pre-processing methods for a fair comparison. We introduce a novel architecture and method for high-fidelity waveform generation without any pre-processing.

[2]We observed that using band splitting of MBD without EQ processor results in white noise on the generated sample in our preliminary study so we introduce discrete wavelet Transform based multi-band modeling.

waveform generation task. We also found that the skip features of our model contain a large ratio of high-frequency information. Additionally, this also provided the noisy high-frequency information to the UBlock at the initial sampling steps. Hence, the accumulated high-frequency noise prevents modeling the high-frequency information of waveform. To reduce this issue, we adopt FreeU by scaling down the skip features $z_{skip}$ and scaling up the backbone features $x$ as follows:

$$x = \alpha \cdot z_{skip} + \beta \cdot x \tag{5}$$

where we found the optimal hyper-parameters through grid search: $\alpha = 0.9$ and $\beta = 1.1$ at the Table 25, and this significantly improve the high-frequency modeling performance in terms of spectral distances. We also found that scaling up the backbone features could improve the perceptual quality by reducing the noisy sound which is included in ground-truth Mel-spectrogram.

## 4 EXPERIMENT AND RESULT

### 4.1 EXPERIMENTAL SETUP

**Dataset**   We train the models using LJSpeech (Ito & Johnson, 2017) and LibriTTS (Zen et al., 2019) datasets. LJSpeech is a high-quality single-speaker dataset with a sampling rate of 22,050 Hz. LibriTTS is a multi-speaker dataset with a sampling rate of 24,000 Hz. Following (Lee et al., 2023), we adopt the same configuration for Mel-spectrogram transformation. For the LJSpeech, we use the Mel-spectrogram of 80 bins. For the LibriTTS, we utilize the Mel-spectrogram of 100 bins.

**Training**   For the LibriTTS dataset, we train PeriodWave using the AdamW optimizer with a learning rate of $2 \times 10^{-4}$, batch size of 128 for 1M steps on four NVIDIA A100 GPUs. Each band of PeriodWave-MB is trained using the AdamW optimizer with a learning rate of $2 \times 10^{-4}$, batch size of 64 for 1M steps on two NVIDIA A100 GPUs.[3] It only takes three days to train the model while GAN-based models take over three weeks. We do not apply any learning rate schedule. For the ablation study, we train the model with a batch size of 128 for 0.5M steps on four NVIDIA A100 GPUs.

**Sampling**   For the ODE sampling, we utilize Midpoint methods with sampling steps of 16.[4] Additionally, we compared the ODE methods including Euler, Midpoint, and RK4 methods according to different sampling steps in Appendix D. The experimental details are described in Appendix J and K.

Table 1: Objective evaluation results on LJSpeech. We utilized the official checkpoints for all models. BigVGAN♥ models are trained with LJSpeech, VCTK, and LibriTTS datasets.

| Method | Training Steps | Params (M) | M-STFT (↓) | PESQ (↑) | Periodicity (↓) | V/UV F1 (↑) | Pitch (↓) | UTMOS (↑) |
|---|---|---|---|---|---|---|---|---|
| Ground Truth | - | - | - | - | - | - | - | 4.3804 |
| HiFi-GAN (V1) | 2.5M | 14.01 | 1.0341 | 3.646 | 0.1064 | 0.9584 | 26.839 | 4.2691 |
| BigVGAN-base♥ | 5.0M | 14.01 | 1.0046 | 3.868 | 0.1054 | 0.9597 | 25.142 | 4.1986 |
| BigVGAN♥ | 5.0M | 112.4 | **0.9369** | 4.210 | 0.0782 | 0.9713 | 19.019 | 4.2172 |
| PriorGrad (50 steps) | 3.0M | 2.61 | 1.2784 | 3.918 | 0.0879 | 0.9661 | 17.728 | 3.6282 |
| FreGrad (50 steps) | 1.0M | 1.78 | 1.2913 | 3.275 | 0.1302 | 0.9490 | 27.317 | 3.1522 |
| PeriodWave-MB (16 steps) | 0.5M | 37.08×2 | 1.1722 | 4.276 | **0.0701** | **0.9730** | 15.143 | 4.2940 |
| PeriodWave (16 steps) | 1.0M | 29.73 | 1.1464 | 4.288 | 0.0744 | 0.9704 | **15.042** | 4.3243 |
| PeriodWave+FreeU (16 steps) | 1.0M | 29.73 | 1.1132 | **4.293** | 0.0749 | 0.9701 | 15.753 | **4.3578** |

### 4.2 LJSPEECH: HIGH-QUALITY SINGLE SPEAKER DATASET WITH 22,050 HZ

We conducted an objective evaluation to compare the performance of the single-speaker dataset. We utilized the official implementation and checkpoints of HiFi-GAN, PriorGrad, and FreGrad (Nguyen et al., 2024), which have the same Mel-spectrogram configuration. Table 1 shows that our model achieved a significantly improved performance in all objective metrics without M-STFT. Additionally, GAN-based models take much more time to train the model due to the discriminators. Furthermore, our proposed methods require smaller sampling steps than diffusion-based models. We observed that diffusion-based model and flow matching-based models could not model the high-frequency information because their objective function does not guarantee the high-frequency information while GAN-based models utilize Mel-spectrogram loss and M-STFT-based discriminators. To reduce this issue, we utilize multi-band modeling and FreeU operation, and the results also show improved performance in most metrics.

---

[3]Due to the limited resources, we only used two GPUs for each band.

[4]The results in Appendix D show that increasing sampling steps improve the performance consistently.

Table 2: Objective and subjective evaluation results on LibriTTS. Following BigVGAN (Lee et al., 2023), objective results are obtained from LibriTTS-dev subsets, and subjective results are obtained from LibriTTS-test subsets. We included the objective metrics of models[†] reported by BigVGAN. For MOS and Pitch, we utilize the official checkpoints of all models without UnivNet. Note that BigVGAN-base and BigVGAN are trained for 5M steps while our models are trained for 1M steps.

| Method | Params (M) | M-STFT (↓) | PESQ (↑) | Periodicity (↓) | V/UV F1 (↑) | Pitch (↓) | MOS (↑) |
|---|---|---|---|---|---|---|---|
| Ground Truth | - | - | - | - | - | - | 3.94±0.03 |
| WaveGlow-256[†] | 99.43 | 1.3099 | 3.138 | 0.1485 | 0.9378 | - | - |
| WaveFlow-128[†] | 22.58 | 1.1120 | 3.027 | 0.1416 | 0.9410 | - | - |
| HiFi-GAN (V1)[†] | 14.01 | 1.0017 | 2.947 | 0.1565 | 0.9300 | - | - |
| UnivNet-c32 | 14.87 | 0.8947 | 3.284 | 0.1305 | 0.9347 | 53.021 | 3.91±0.03 |
| Vocos | 13.53 | 0.8544 | 3.615 | 0.1113 | 0.9470 | 24.075 | 3.89±0.03 |
| BigVGAN-base[†] | 14.01 | 0.8788 | 3.519 | 0.1287 | 0.9459 | 24.432 | 3.91±0.03 |
| BigVGAN[†] | 112.4 | **0.7997** | 4.027 | 0.1018 | 0.9598 | 25.651 | 3.92±0.03 |
| PeriodWave-MB (16 steps) | 37.08×4 | 0.9729 | **4.262** | **0.0704** | **0.9678** | **16.829** | **3.95±0.03** |
| PeriodWave (16 steps) | 29.80 | 1.2129 | 4.224 | 0.0762 | 0.9652 | 18.730 | 3.93±0.03 |
| PeriodWave + FreeU (16 steps) | 29.80 | 1.0269 | 4.248 | 0.0765 | 0.9651 | 17.398 | **3.95±0.03** |

Table 3: Objective evaluation results with different training steps on LibriTTS dataset.

| Methods | Training Steps | M-STFT (↓) | PESQ (↑) | Periodicity (↓) | V/UV F1 (↑) | UTMOS (↑) |
|---|---|---|---|---|---|---|
| PeriodWave-MB (16 steps) | 1M | 0.9729 | 4.262 | 0.0704 | 0.9678 | 3.6534 |
| | 0.5M | 0.9932 | 4.213 | 0.0745 | 0.9653 | 3.6142 |
| | 0.3M | 1.0697 | 4.161 | 0.0777 | 0.9640 | 3.5641 |
| | 0.15M | 1.1003 | 4.020 | 0.0842 | 0.9580 | 3.4983 |

## 4.3 LIBRITTS: MULTI-SPEAKER DATASET WITH 24,000 HZ

We conducted objective and subjective evaluations to compare the performance of the multi-speaker dataset. We utilized the publicly available checkpoints of UnivNet, BigVGAN, and Vocos, which are trained with the LibriTTS dataset. Table 24 shows our model significantly improved performance in all metrics but the M-STFT metric. Although other GAN-based models utilize Mel-spectrogram distance loss and multi-resolution spectrogram discriminators which can minimize the distance on the spectral domain, we only trained the model by minimizing the distance of the vector field on the waveform. However, our model achieved better performance in subjective evaluation. Specifically, our models have better performance on the periodicity metrics, and this means that our period-aware structure could improve the performance in terms of pitch and periodicity by significantly reducing the jitter sound. Both PeriodWave-MB and PeriodWave demonstrated significantly lower pitch error distances compared to BigVGAN. Specifically, PeriodWave-MB and PeriodWave (FreeU) achieved a pitch error distance of 16.829 and 18.730 (17.398), respectively, while BigVGAN's pitch error distance was 25.651. Table 3 also demonstrated the fast training speed of PeriodWave. The model trained for 0.15M steps could achieve comparable performance compared to baseline models which are trained over 1M steps.

Table 4: Objective evaluation results with different temperature $\tau$.

| Methods | Temperature $\tau$ | M-STFT (↓) | PESQ (↑) | Periodicity (↓) | V/UV F1 (↑) | UTMOS (↑) |
|---|---|---|---|---|---|---|
| PeriodWave-MB | 1.0 | **0.9363** | 4.152 | 0.0721 | **0.9679** | 3.5194 |
| | **0.667** | 0.9729 | 4.262 | **0.0704** | 0.9678 | **3.6534** |
| | 0.333 | 1.0915 | **4.278** | 0.0729 | 0.9668 | 3.5457 |
| | 0.1 | 1.3062 | 3.847 | 0.0788 | 0.9634 | 3.1442 |

## 4.4 SAMPLING ROBUSTNESS, DIVERSITY, AND CONTROLLABILITY

We utilize a flow matching model for PeriodWave, allowing it to generate diverse samples with different Gaussian noise. However, our goal is a conditional generation using the Mel-spectrogram. We need to decrease the diversity to improve the robustness of the model. To achieve this, we can multiply the small scale of temperature $\tau$ to the Gaussian noise during inference. Table 4 shows that using $\tau$ of 0.667 could improve the performance. We also observed that samples generated with a $\tau$ of 1.0 contain a small amount of white noise, which decreases perceptual quality despite having the lowest lowest M-STFT metrics. Furthermore, we could control the energy for each band by using different scales of $\tau$. This approach could be utilized for a neural EQ that can generate the signal by reflecting the conditioned energy, not merely manipulating the energy of the generated samples.

Table 5: Objective evaluation results on out-of-distribution samples from MUSDB18-HQ. We evaluated Periodicity, and V/UV F1 on vocal samples from MUSDB18-HQ.

| Method | M-STFT (↓) | PESQ (↑) | Periodicity (↓) | V/UV F1 (↑) |
|---|---|---|---|---|
| UnivNet-c32 | 1.1377 | 1.678 | 0.1588 | 0.9186 |
| Vocos | 1.0203 | 2.173 | 0.1305 | 0.9454 |
| BigGAN-base | 1.0132 | 2.315 | 0.1272 | 0.9307 |
| BigVGAN | **0.9062** | 2.862 | 0.0959 | 0.9501 |
| PeriodWave-MB (16 steps) | 1.0490 | **3.120** | **0.0945** | **0.9524** |
| PeriodWave (16 steps, Midpoint) | 1.2702 | 2.959 | 0.1046 | 0.9475 |
| PeriodWave + FreeU (16 steps, Midpoint) | 1.1923 | 3.062 | 0.0994 | 0.9479 |

Table 6: 5-scale SMOS results on out-of-distribution samples from MUSDB18-HQ.

| Method | Vocal | Drums | Bass | Others | Mixture | Average |
|---|---|---|---|---|---|---|
| Ground Truth | 3.85±0.11 | 4.00±0.11 | 3.83±0.11 | 4.01±0.11 | 4.03±0.10 | 3.94±0.05 |
| UnivNet-c32 | 3.32±0.15 | 3.40±0.16 | 2.89±0.16 | 2.92±0.18 | 2.80±0.15 | 3.06±0.07 |
| Vocos | 3.57±0.12 | 3.64±0.13 | 2.89±0.16 | 3.21±0.17 | 3.16±0.13 | 3.29±0.06 |
| BigVGAN-base | 3.64±0.13 | 3.68±0.13 | 3.07±0.14 | 3.31±0.15 | 3.51±0.13 | 3.44±0.06 |
| BigVGAN | 3.63±0.12 | **4.01±0.12** | 3.13±0.13 | 3.53±0.15 | **3.56±0.13** | 3.56±0.06 |
| PeriodWave (16 steps) | 3.70±0.12 | 3.76±0.14 | 3.20±0.15 | 3.38±0.13 | 3.44±0.13 | 3.50±0.06 |
| PeriodWave-MB (16 steps) | **3.72±0.12** | 3.71±0.13 | **3.52±0.13** | **3.72±0.14** | 3.51±0.13 | **3.63±0.06** |

## 4.5 MUSDB18-HQ: MULTI-TRACK MUSIC AUDIO DATASET FOR OOD ROBUSTNESS

To evaluate the robustness on the out-of-distribution samples, we measure performance on the MUSDB18-HQ dataset that consists of multi-track music audio including vocals, drums, bass, others, and a mixture. We utilize all test samples including 50 songs with 5 tracks, and randomly sample the 10-second segments for each sample. Table 5 shows our model has better performance on all metrics without M-STFT. Table 6 shows that PeriodWave-MB outperformed the baseline models by improving the out-of-distribution robustness. Specifically, we significantly improve the performance of bass, the frequency range of which is known between 40 to 400 Hz. Additionally, we observed that our model significantly reduces the jitter sound in the out-of-distribution samples.

Table 7: Ablation study on LibriTTS. All models are trained for 0.5M steps.

| Method | Period | M-STFT (↓) | PESQ (↑) | Periodicity (↓) | V/UV F1 (↑) | UTMOS (↑) |
|---|---|---|---|---|---|---|
| Ground Truth | - | - | - | - | - | 3.8626 |
| PeriodWave-MB | [1,2,3,5,7] | 0.9932 | 4.213 | 0.0745 | 0.9653 | 3.6142 |
| PeriodWave | [1,2,3,5,7] | 1.1737 | 4.072 | 0.0806 | 0.9627 | 3.5544 |
| PeriodWave w/o Prior | [1,2,3,5,7] | 1.3754 | 3.900 | 0.0930 | 0.9562 | 3.5352 |
| PeriodWave w/o Mel Encoder | [1,2,3,5,7] | 1.5194 | 2.511 | 0.1093 | 0.9457 | 2.6737 |
| PeriodWave | [1] | 1.2588 | 3.795 | 0.0885 | 0.9572 | 3.4215 |
| | [1,1,1,1,1] | 1.1337 | 3.964 | 0.0888 | 0.9597 | 3.4728 |
| | [1,1,1,3,3] | 1.1234 | 4.011 | 0.0818 | 0.9643 | 3.4879 |
| | [1,1,1,3,9] | 1.2736 | 4.061 | 0.0830 | 0.9644 | 3.5057 |
| | [1,2,4,6,8] | 1.1481 | 4.075 | 0.0782 | 0.9647 | 3.5468 |
| | [1,2,4,8,16] | 1.1463 | 4.124 | 0.0787 | 0.9639 | 3.5408 |
| | [1,2,3,5,7,11,13,17] | 1.1617 | 4.125 | 0.0792 | 0.9610 | 3.5384 |

## 4.6 ABLATION STUDY

**Different Periods** We conduct ablation study for different periods at the same structure. Table 7 shows that the model with a period of 1 shows the lowest performance. Increasing the number of periods could improve the entire performance in terms of most metrics, consistently. However, this also improves the computational cost and requires more training steps for optimizing various periods in a single estimator so we fix the model with the period of [1,2,3,5,7]. Meanwhile, we compared the model with periods of [1,2,4,6,8] and [1,2,4,8,16] to demonstrate the effectiveness of the prime number for the period. We observed that using prime number could improve the UTMOS slightly and the model with periods of [1,2,4,6,8] and [1,2,4,8,16] also have comparable performance, which can reflect the different period representations of the waveform. We thought that the model with periods of [1,2,3,5,7,11,13,17] requires more training steps. The model with periods of [1,1,1,1,1] showed slightly better performance than the model with a periods of [1]. However, it significantly underperforms compared to other models using different size of periods. The models with periods of [1,1,1,3,3] and [1,1,1,3,9] have better performance than [1,1,1,1,1], and the results demonstrate that using different periods could perform better.This also demonstrates that our new waveform generator structure is suitable for waveform generation.

**Prior**  PriorGrad demonstrated that data-dependent prior information could improve the performance and sampling speed for diffusion models. We also utilize the normalized energy which can be extracted Mel-spectrogram as prior information. We observe that the data-dependent prior could improve the quality and sampling speed in flow matching based models. Meanwhile, although we failed to implement the quality reported by SpecGrad (Koizumi et al., 2022), we see that the spectrogram-based prior could improve the performance rather than the energy-based prior.

**Mel Encoder**  Our Mel encoder significantly improved the performance through efficient time-frequency modeling. This only requires a small increase in computation cost because we reused the extracted features which are fed to the period-aware flow matching estimator for each sampling step.

Table 8: Comparison of Diffusion and CFM. All models are trained for 1M steps.

| Method | Steps | M-STFT (↓) | PESQ (↑) | Periodicity (↓) | V/UV F1 (↑) | Pitch (↓) | UTMOS (↑) |
|---|---|---|---|---|---|---|---|
| PeriodWave w/ CFM | 32 | 1.072 | 4.233 | 0.078 | 0.964 | 17.418 | 3.646 |
| PeriodWave w/ CFM | 25 | 1.159 | 4.233 | 0.078 | 0.964 | 17.420 | 3.650 |
| PeriodWave w/ CFM | 16 | 1.212 | 4.224 | 0.076 | 0.965 | 17.496 | 3.649 |
| PeriodWave w/ CFM | 6 | 1.379 | 4.178 | 0.082 | 0.959 | 23.223 | 3.628 |
| PeriodWave w/ DDPM | 50 | 1.159 | 4.151 | 0.084 | 0.961 | 23.046 | 3.377 |
| PeriodWave w/ DDPM | 6 | 1.233 | 3.541 | 0.095 | 0.958 | 24.351 | 2.953 |

**CFM vs Diffusion**  We compared the model trained with diffusion and CFM to demonstrate the effectiveness of CFM for waveform generation. For a fair comparison, we utilize the same model architecture. Furthermore, we utilize a prior-based diffusion models following PriorGrad (Lee et al., 2022b). We utilize the same noise scheduling method of PriorGrad. For time steps of 6, we utilized noise scheduling of $[0.0001, 0.008, 0.010.05, 0.7, 0.9]$. Table 8 shows that CFM outperformed diffusion in terms of all metrics. Furthermore, this indicates that CFM has several advantages in performance and efficiency where the model with CFM has better performance even with a smaller sampling steps than the model trained with DDPM. Furthermore, our model was trained with a continuous time t while DDPM should be trained with discrete time t. We observed that training the diffusion-based model for continuous time t requires much more training times to optimize the models. However, while our model was trained with a continuous time t, our models show a faster convergence speed in that Table 8 indicated the effectiveness of CFM compared to diffusion.

### 4.7 MULTI SPEAKER TEXT-TO-SPEECH

We conduct two-stage multi-speaker TTS experiments to further demonstrate the robustness of the proposed models compared to previous large-scale GAN-based models including BigVGAN and BigVSAN (Shibuya et al., 2024). Note that BigVGAN and BigVSAN were trained for 5M and 10M steps, respectively. We utilize ARDiT-

Table 9: Zero-shot TTS Results. We utilized ARDiT-TTS trained with LibriTTS as TTS model.

| Methods | MOS (↑) | UTMOS (↑) |
|---|---|---|
| BigVSAN | 3.99±0.01 | 3.9732 |
| BigVGAN | 4.03±0.01 | 4.0424 |
| PeriodWave (16 steps) | 4.06±0.01 | 4.2209 |
| PeriodWave + FreeU (16 steps) | **4.07±0.01** | **4.2621** |

TTS (Liu et al., 2024b) as zero-shot TTS model which was trained with LibriTTS dataset. We convert 500 samples of generated Mel-spectrogram into waveform signal by each model. The Table 9 shows that our model has better performance on the objective and subjective metrics in terms of UTMOS and MOS. Furthermore, Our model with FreeU has much better performance than others. We can discuss that FreeU could reduce the high-frequency noise resulting in better perceptual quality.

We also discussed the train-inference mismatch problem of two-stage TTS and the effectiveness of iterative refinement to reduce this problem and added single-speaker TTS results in Appendix G.

### 4.8 LIMITATION

Although our models could generate the waveform with small sampling steps, Table E shows that our models have a slow synthesis speed compared to GAN-based models. To overcome this issue, we will explore distillation methods or adversarial training to reduce the sampling steps for much more fast inference by using our period-aware structure. Additionally, our models still show a lack of robustness in terms of high-frequency information because we only train the model by estimating the vector fields on the waveform resolution. Although multi-band modeling reduced this issue, it requires more complex model designs. So, we have a plan to add a modified spectral objective function or blocks that can reflect the spectral representations when estimating vector fields by utilizing short-time Fourier convolution proposed in (Han & Lee, 2022) for audio super-resolution. Moreover, we see that classifier-free guidance could be adapted to our model to improve the audio quality.

Table 10: Objective evaluation results of parallel generation from discrete token.

| Method | Params (M) | CER (↓) | WER (↓) | M-STFT (↓) | PESQ (↑) | Periodicity (↓) | V/UV F1 (↑) | Pitch (↓) | UTMOS (↑) |
|---|---|---|---|---|---|---|---|---|---|
| Ground Truth | - | 0.92 | 2.94 | - | - | - | - | - | 3.8626 |
| Mimi ($Q = 32$) | 79.30 | 1.28 | 3.81 | 1.1067 | 3.469 | 0.0886 | 0.9541 | 28.084 | 3.8005 |
| Mimi ($Q = 16$) | 79.30 | 1.35 | 3.81 | 1.2119 | 2.911 | 0.1232 | 0.9345 | 34.863 | 3.7455 |
| Mimi ($Q = 8$) | 79.30 | 3.07 | 6.91 | 1.3509 | 2.266 | 0.1651 | 0.9104 | 50.679 | 3.5068 |
| PeriodWave ($Q = 8$, 2 steps) | 35.86 | 2.50 | 5.56 | 1.2651 | **2.293** | **0.1413** | **0.9244** | 44.889 | 3.8827 |
| PeriodWave ($Q = 8$, 4 steps) | 35.86 | **2.40** | **5.45** | **1.2432** | 2.256 | 0.1424 | 0.9234 | **44.691** | **3.9343** |

Table 11: Objective evaluation results of streaming generation from discrete token.

| Method | Params (M) | CER (↓) | WER (↓) | M-STFT (↓) | PESQ (↑) | Periodicity (↓) | V/UV F1 (↑) | Pitch (↓) | UTMOS (↑) |
|---|---|---|---|---|---|---|---|---|---|
| Mimi ($Q = 8$) | 79.30 | 3.05 | 6.93 | 1.3522 | **2.266** | 0.1650 | 0.9104 | 50.686 | 3.5068 |
| PeriodWave ($Q = 8$, 2 steps) | 35.86 | **2.45** | **5.53** | **1.2716** | 2.233 | **0.1429** | **0.9228** | **40.634** | **3.8508** |

# 5 AUDIO GENERATION FROM DISCRETE TOKEN

We conducted experiments for parallel and streaming generation from discrete tokens. We used Mimi of Moshi (Défossez et al.), a state-of-the-art neural audio codec that operates at 12.5 Hz. Note that the number of codebooks in Mimi can be up to 32, but Moshi utilized a $Q = 8$ quantizer for speech language models, so we also used the same discrete tokens from the eight quantizer as an input to our model instead of Mel-spectrogram. Specifically, we used the post-quantized latent representation as input for PeriodWave. For streaming generation, we introduce a single token delayed generation. Although the model was trained for parallel generation, the samples with streaming generation show minimal degradation. The details of streaming generation are described in Appendix I and Figure 4.

We trained the model using segments of 48,000 frames (2s waveform and 25 token embeddings). We modified the Mel-encoder by adding an additional upsampling layer to upsample a high-compressed codec at 12.5 Hz to waveform signal with sampling rate of 24,000 Hz. Since energy can not be extracted from discrete tokens, we remove the energy-based prior. Furthermore, we fine-tuned Period-Wave by fixing the number of iteration steps to accelerate the inference speed for streaming generation, and utilized an adversarial training to improve the performance without using reconstruction loss.

We first conducted CER and WER evaluations using Whisper-large-v3-turbo (Radford et al., 2023) to evaluate the pronunciation and semantic consistency of generated samples from low-bitrate discrete tokens. Table 10 shows that our model can generate speech with better pronunciation by preserving semantic information. Also, our models with $Q = 8$ quantizer have better naturalness than Mimi with $Q = 32$ quantizer in terms of UTMOS. PeriodWave can enhance the audio quality by increasing the iteration steps in parallel generation. Furthermore, Table 11 demonstrates our proposed streaming generation can synthesize the speech in a streaming manner with a minimal degradation. Additional experiments of streaming generation are reported in Appendix I and Table 20.

We believe that our model can be used for alternative decoder of neural audio codec models. We see that our model with parallel generation can facilitate the high-quality dialogue data collection from speech language models thanks to enhanced audio quality. However, there is room for improvement of efficiency in streaming audio generation. In future, we have a plan to fine-tune the model with chucked auto-regressive or masking-infilling for in-context streaming generation.

# 6 CONCLUSION

In this work, we proposed PeriodWave, a novel universal waveform generation model with conditional flow matching. Motivated by the multiple periodic characteristics of high-resolution waveform signals, we introduce the period-aware flow matching estimators which can reflect different implicit periodic representations when estimating vector fields. Furthermore, we observed that increasing the number of periods improve the performance, and we introduce a period-conditional universal estimator for efficient structure. By adopting this, we also implement a period-wise batch inference for efficient inference. The experimental results demonstrate the superiority of our model in high-quality waveform generation and OOD robustness. Furthermore, we demonstrate the effectiveness of proposed model in neural audio codec decoding tasks both in parallel and streaming generation. GAN-based models still hold great potential and have shown strong performance but require multiple loss functions, resulting in complex training and long training times. On the other hand, we introduced a flow matching based approach using a single loss function, which offers notable advantages in both efficiency and performance.

We released all source code and checkpoints, hoping that our approach will facilitate research in waveform generation. We believe that its efficiency and flexibility make it an ideal backbone, easily adaptable to domain-specific or personalized data, and fine-tuning with adversarial training.

## ACKNOWLEDGEMENT

We'd like to thank Yeongtae Hwang for helpful discussion and contributions to our work suggesting FreeU to address a high-frequency error of diffusion models. We sincerely thank Zhijun Liu, the author of ARDiT-TTS for providing the Mel spectrograms of ARDiT-TTS, which enabled us to perform the second stage of TTS synthesis. This work was supported by Institute of Information & communications Technology Planning & Evaluation (IITP) grant funded by the Korea government (MSIT) (IITP-2025-RS-2023-00255968, the Artificial Intelligence Convergence Innovation Human Resources Development and No. 2021-0-02068, Artificial Intelligence Innovation Hub) and Artificial intelligence industrial convergence cluster development project funded by the Ministry of Science and ICT(MSIT, Korea)&Gwangju Metropolitan City.

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

## A   FLOW MATCHING WITH OPTIMAL TRANSPORT PATH

In the data space $\mathbb{R}^d$, let us consider an observation $x \in \mathbb{R}^d$ sampled from an unknown distribution $q(x)$. Continuous Normalizing Flows (CNFs) transform a simple prior $p_0$ into a target distribution $p_1 \approx q$ using a time-dependent vector field $v_t : [0, 1] \times \mathbb{R}^d \to \mathbb{R}^d$. The flow $\phi_t : [0, 1] \times \mathbb{R}^d \to \mathbb{R}^d$ is defined by the ordinary differential equation:

$$\frac{d}{dt}\phi_t(x) = v_t(\phi_t(x); \theta), \quad \phi_0(x) = x, \quad x \sim p_0, \tag{6}$$

where $\phi_t(x)$ denotes the state of the system at time $t$, driven by the vector field $v_t(\cdot; \theta)$. The probability density path $p_t : [0, 1] \times \mathbb{R}^d \to \mathbb{R}_{>0}$ of this flow can be derived using the change of variables. Specifically, this system transforms the initial probability density $p_0$ to $p_t$ at time $t$, and the resulting probability density $p_t$ is given by:

$$p_t(y) = p_0(\phi_t^{-1}(y)) \left| \det\left( \frac{\partial \phi_t^{-1}}{\partial y} \right) \right|. \tag{7}$$

Given samples from an unknown data distribution $q(x)$, our objective is to transform a simple initial distribution $p_0$ (e.g., standard normal) into a target distribution $p_1 \approx q$. The challenge lies in the fact that both $p_t$ and the corresponding vector field $u_t$ that generates $p_t$ are generally unknown.

To address this, (Lipman et al., 2022) introduce the flow matching objective, which aims to match the vector field $v_t(x)$ to an ideal vector field $u_t(x)$ that would generate the desired probability path $p_t$. The flow matching training objective involves minimizing the loss function $L_{FM}(\theta)$, which is defined by regressing the model's vector field $v_\theta(t, x)$ to a target vector field $u_t(x)$ as follows:

$$\mathcal{L}_{FM}(\theta) = \mathbb{E}_{t \sim [0,1], x \sim p_t(x)} ||v_\theta(t, x) - u_t(x)||_2^2. \tag{8}$$

Here, $t \sim U[0, 1]$, and $v_t(x; \theta)$ is a neural network with parameters $\theta$. However, direct access to $u_t$ and $p_t$ is challenging, prompting the introduction of Conditional Flow Matching (CFM):

$$\mathcal{L}_{CFM}(\theta) = \mathbb{E}_{t \sim [0,1], x \sim p_t(x|z)} ||v_\theta(t, x) - u_t(x|z)||_2^2. \tag{9}$$

This expression replaces the impractical marginal probability density and vector field with conditional probability density and conditional vector field, enabling a feasible approach. Crucially, $L_{\text{CFM}}(\theta)$ and $L_{\text{FM}}(\theta)$ share identical gradients with respect to $\theta$, ensuring equivalent efficacy in model training. The probability path $p_t(x)$ and the associated vector field $u_t(x)$ can be expressed conditionally as follows:

$$p_t(x) = \int p_t(x|z)p(z)dz \quad \text{and} \quad u_t(x) = \int \frac{p_t(x|z)u_t(x|z)}{p_t(x)} p(z)dz, \tag{10}$$

where $p(z)$ is an arbitrary conditional distribution independent of $x$ and $t$. Assuming the existence of an optimal vector field $u_t$, the neural network $v_t(x; \theta)$ can learn this vector field. Furthermore, (Lipman et al., 2022) indicates that conditional vector field estimation is equivalent to unconditional vector field estimation:

$$\min_\theta \mathbb{E}_{t, p_t(x)} ||u_t(x) - v_t(x; \theta)||^2 \equiv \min_\theta \mathbb{E}_{t, q(x_1), p_t(x|x_1)} ||u_t(x \mid x_1) - v_t(x; \theta)||^2, \tag{11}$$

where $p_0(x \mid x_1) = p_0(x)$ and $p_1(x \mid x_1) = N(x \mid x_1, \sigma^2 I)$ assume sufficiently small $\sigma$. Lastly, generalizing this technique with the noise condition $x_0 \sim N(0, 1)$, we consider the OT-CFM loss as follows:

$$L_{\text{OT-CFM}}(\theta) = \mathbb{E}_{t, q(x_1), p_0(x_0)} ||u_t^{\text{OT}}(\phi_t^{\text{OT}}(x_0) \mid x_1) - v_t(\phi_t^{\text{OT}}(x_0) \mid \mu; \theta)||^2, \tag{12}$$

where $\mu$ is the frame-wise predicted mean of $x_1$, and $\phi_t^{\text{OT}}(x_0) = (1 - (1 - \sigma_{\min})t)x_0 + tx_1$ represents the flow from $x_0$ to $x_1$. The target conditional vector field $u_t^{\text{OT}}(\phi_t^{\text{OT}}(x_0) \mid x_1) = x_1 - (1 - \sigma_{\min})x_0$ enhances performance due to its inherent linearity. This approach efficiently manages data transformation and significantly enhances training speed and efficiency by integrating optimal transport paths.

# B Implementation Details

For reproducibility, we will release all source code, checkpoints, and generated samples at `https://periodwave.github.io/demo/`. We also describe the hyperparameter details of our models at Table 12.

Table 12: Hyperparameters of PeriodWave.

| Module | Hyperparameter | PeriodWave | PeriodWave-MB |
|---|---|---|---|
| Period-aware FM Estimator (UNet) | Downsampling Ratio | [1,4,4,4] | [1,4,4,1] |
| | Upsampling Ratio | [4,4,4] | [4,4,1]] |
| | DBlock Hidden Dim | [32,64,128] | [32,128,512] |
| | MBlock Hidden Dim | 512 | 512 |
| | UBlock Hidden Dim | [128,64,32] | [512,128,32] |
| | ResBlock Kernel Size | [3,3] | [3,3] |
| | ResBlock Dilation Size | [1,2] | [1,2]] |
| | Period | [1,2,3,5,7] | [1,2,3,5,7] |
| | Activation | SiLU | SiLU |
| | Final ResBlock Kernel Size | [3,3,3] | [3,3,3] |
| | Final ResBlock Dilation Size | [1,2,4] | [1,2,4] |
| Cond. Layer | Time Embedding | 256 | 256 |
| | Period Embedding | 256 | 256 |
| | MLP | [512, 2048, 512] | [512, 2048, 512] |
| Mel Encoder | Mel Embedding | 512 | 512 |
| | First ConvNext V2 Blocks | 8 | 8 |
| | Hidden Dim | 1536 | 1536 |
| | Drop Path | 0.1 | 0.1 |
| | Upsampling Ratio | 4 | 4 |
| | Upsampling Dim | 256 | 256 |
| | Second ConvNext V2 Blocks | 4 | 4 |
| | Second Hidden Dim | 1024 | 1024 |
| | Downsampling ratio | [1,2,3,5,7] | [1,2,3,5,7] |
| | Output Dim | 512 | 512 |
| Energy-based Prior | Full-band Energy Max/Min | 9.124346/0.031622782 | - |
| | First-band Energy Max/Min | - | 8.756637/0.024698181 |
| | First-band Start/End Bin | - | [0:61] |
| | Second-band Energy Max/Min | - | 4.242267/0.014491379 |
| | Second-band Start/End Bin | - | [60:81] |
| | Third-band Energy Max/Min | - | 3.1011465/0.011401756 |
| | Third-band Start/End Bin | - | [80:93] |
| | Fourth-band Energy Max/Min | - | 2.3407087/0.031622782 |
| | Fourth-band Start/End Bin | - | [91:100] |
| Mel-spectrogram | FFT Size | 1024 | 1024 |
| | Hop Size | 256 | 256 |
| | Window Size | 1024 | 1024 |
| | Bins | 100 | 100 |
| | F0 Min/Max | 0/12000 | 0/12000 |
| Others | Training Step | 1M | 1M |
| | Learning Rate | $2 \times 10^{-4}$ | $2 \times 10^{-4}$ |
| | Learning Scheduling | - | - |
| | Batch Size | 128 | 64 |
| | GPUs | 4 | 2 |
| | Noise Scale $\alpha$ | 0.5 | 0.5 |
| | Segment Size | 32,768 | 32,768 |
| | Temperature $\tau$ | 0.667 | 0.667 |
| | ODE Sampling Steps | 16 | 16 |
| | $s_w$ | 0.9 | - |
| | $b_w$ | 1.1 | - |

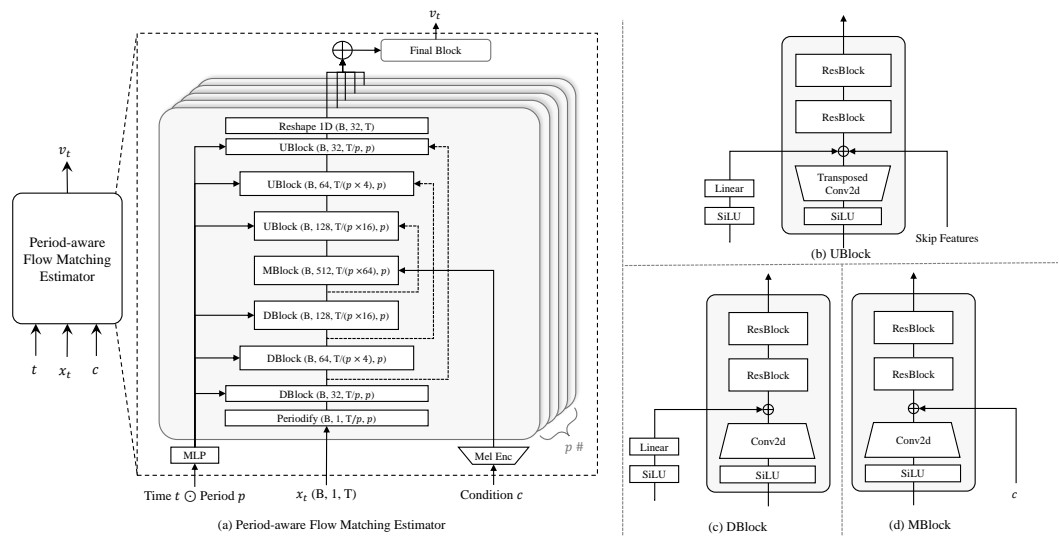

Figure 3: Architecture of PeriodWave

Table 13: Objective evaluation results on LJSpeech. We utilized the official checkpoints for all models. BigVGAN♥ models are trained with LJSpeech, VCTK, and LibriTTS datasets.

| Method | Training Steps | Params (M) | M-STFT (↓) | PESQ (↑) | Periodicity (↓) | V/UV F1 (↑) | Pitch (↓) | UTMOS (↑) |
|---|---|---|---|---|---|---|---|---|
| Ground Truth | - | - | - | - | - | - | - | 4.3804 |
| HiFi-GAN (V1) | 2.50M | 14.01 | 1.0341 | 3.646 | 0.1064 | 0.9584 | 26.839 | 4.2691 |
| BigVGAN-base♥ | 5.00M | 14.01 | 1.0046 | 3.868 | 0.1054 | 0.9597 | 25.142 | 4.1986 |
| BigVGAN♥ | 5.00M | 112.4 | 0.9369 | 4.210 | 0.0782 | 0.9713 | 19.019 | 4.2172 |
| PriorGrad (50 steps) | 3.00M | 2.61 | 1.2784 | 3.918 | 0.0879 | 0.9661 | 17.728 | 3.6282 |
| FreGrad (50 steps) | 1.00M | 1.78 | 1.2913 | 3.275 | 0.1302 | 0.9490 | 27.317 | 3.1522 |
| PeriodWave-MB | 0.05M | 37.08×2 | 1.2048 | 3.785 | 0.0873 | 0.9641 | 18.050 | 4.0662 |
| (16 steps) | 0.15M | 37.15×2 | 1.2430 | 4.141 | 0.0759 | 0.9699 | 16.366 | 4.2218 |
|  | 0.30M | 37.08×2 | 1.1574 | 4.246 | 0.0722 | 0.9726 | **14.426** | 4.2788 |
|  | 0.50M | 37.08×2 | 1.1722 | 4.276 | 0.0701 | 0.9730 | 15.143 | 4.2940 |
| PeriodWave | 0.05M | 29.73 | 1.2146 | 3.821 | 0.0982 | 0.9594 | 19.512 | 3.9935 |
| (16 steps) | 0.15M | 29.73 | 1.2112 | 4.144 | 0.0865 | 0.9644 | 17.056 | 4.2198 |
|  | 0.30M | 29.73 | 1.2232 | 4.211 | 0.0884 | 0.9641 | 18.899 | 4.2671 |
|  | 0.50M | 29.73 | 1.1574 | 4.310 | 0.0782 | 0.9685 | 16.104 | 4.3106 |
| PeriodWave (1 step) | 1.00M | 29.73 | 1.5367 | 2.733 | 0.1074 | 0.9524 | 19.018 | 3.2725 |
| PeriodWave (2 step) | 1.00M | 29.73 | 1.3033 | 4.050 | 0.0853 | 0.9650 | 15.980 | 4.1528 |
| PeriodWave (4 step) | 1.00M | 29.73 | 1.2529 | 4.226 | 0.0782 | 0.9691 | 15.736 | 4.2825 |
| PeriodWave (8 step) | 1.00M | 29.73 | 1.2222 | 4.269 | 0.0746 | 0.9704 | 14.944 | 4.3229 |
| PeriodWave (16 step) | 1.00M | 29.73 | 1.1464 | 4.288 | 0.0744 | 0.9704 | 15.042 | 4.3243 |
| PeriodWave+FreeU | 1.00M | 29.73 | 1.1132 | 4.293 | 0.0749 | 0.9701 | 15.753 | **4.3578** |

# C  ADDITIONAL RESULTS ON LJSPEECH

We reported additional results on LJSpeech dataset according to the training steps to demonstrate the effectiveness of our proposed methods. 13 showed that our models achieved a high performance even with small training steps. Furthermore, training the model with 0.3M could outperformed the previous powerful GAN-based neural vocoder in all metrics without M-STFT metrics. It is worth noting that our models are only optimized with a single loss while GAN-based methods utilize discriminator loss, feature matching loss, and Mel reconstruction loss to train the model. Furthermore, they require various discriminators to capture the different features to improve the perceptual quality, and this increase the burden to optimize the model, which requires hyper-parameter tuning including weights for each loss, learning rate, and learning rate scheduling methods.

# D ODE METHODS

Table 14: Objective evaluation results with different ODE methods and sampling steps.

| Methods | ODE | steps | M-STFT (↓) | PESQ (↑) | Periodicity (↓) | V/UV F1 (↑) | UTMOS (↑) |
|---|---|---|---|---|---|---|---|
| PeriodWave-MB | Euler | 1 | 1.8995 | 1.437 | 0.1627 | 0.9114 | 1.5102 |
| | | 2 | 1.3598 | 3.263 | 0.0929 | 0.9556 | 2.9470 |
| | | 4 | 1.2365 | 4.060 | 0.0801 | 0.9635 | 3.4160 |
| | | 8 | 1.1817 | 4.207 | 0.0756 | 0.9654 | 3.5664 |
| | | 16 | 1.1183 | 4.256 | 0.0720 | 0.9672 | 3.6209 |
| | | 32 | 1.0499 | 4.265 | 0.0710 | 0.9674 | 3.6482 |
| | | 64 | 1.0004 | 4.269 | 0.7035 | 0.9678 | 3.6565 |
| | | 128 | 0.9601 | 4.266 | 0.0700 | 0.9677 | 3.6587 |
| | | 256 | 0.9311 | 4.264 | 0.0697 | 0.9678 | 3.6593 |
| PeriodWave-MB | Midpoint | 1 | 1.3342 | 3.241 | 0.0954 | 0.9541 | 2.9725 |
| | | 2 | 1.2072 | 4.132 | 0.0759 | 0.9662 | 3.4662 |
| | | 4 | 1.0825 | 4.232 | 0.0732 | 0.9670 | 3.5899 |
| | | 8 | 1.0457 | 4.252 | 0.0703 | 0.9680 | 3.6311 |
| | | 16 | 0.9729 | 4.262 | 0.0704 | 0.9678 | 3.6534 |
| | | 32 | 0.9586 | 4.263 | 0.0700 | 0.9678 | 3.6583 |
| | | 64 | 0.9291 | 4.263 | 0.0694 | 0.9679 | 3.6583 |
| | | 128 | 0.9094 | 4.270 | 0.0694 | 0.9678 | 3.6577 |
| | | 256 | 0.9016 | 4.270 | 0.0693 | 0.9680 | 3.6573 |
| PeriodWave-MB | RK4 | 1 | 3.6066 | 1.080 | 0.2749 | 0.8106 | 1.2563 |
| | | 2 | 2.6118 | 1.482 | 0.1089 | 0.9452 | 1.7786 |
| | | 4 | 1.9222 | 2.315 | 0.0859 | 0.9591 | 2.8721 |
| | | 8 | 1.4860 | 3.150 | 0.0783 | 0.9642 | 3.2758 |
| | | 16 | 1.1990 | 3.766 | 0.0753 | 0.9654 | 3.4513 |
| | | 32 | 1.0303 | 4.084 | 0.0721 | 0.9677 | 3.5204 |
| | | 64 | 0.9438 | 4.148 | 0.0710 | 0.9683 | 3.5416 |
| | | 128 | 0.9238 | 4.134 | 0.0712 | 0.9683 | 3.5396 |
| | | 256 | 0.9311 | 4.123 | 0.0697 | 0.9696 | 3.5350 |

## D.1 ANALYSIS ON DIFFERENT ODE SAMPLING METHODS

We explore the different ODE methods to analyze the sample quality according to the different sampling steps. We utilize three ODE methods including Euler, Midpoint, and RK4 methods. Table 14 shows that increasing the sampling steps could improve the sample quality in most metrics consistently. We observed that RK4 methods have the lowest performance, resulting in white noise on the generated samples. We can discuss it because we predict the vector field directly including the time point t1 for their last order estimation where it is hard to estimate it at the early time steps, resulting in white noise. Meanwhile, Midpoint method show better performance than Euler method, consistently even with half sampling steps which have a similar computational cost with Euler method. In this regard, we fixed the Midpoint method for our ODE method. Additionally, using a small sampling step could achieve the comparable performance than previous methods.

Table 15: Synthesis speed for baseline models.

| Method | HiFi-GAN (V1) | BigVGAN-base | BigVGAN | PriorGrad | FreGrad |
|---|---|---|---|---|---|
| Syn.Speed | 166.70× | 105.18× | 38.28× | 8.42× | 10.88× |
| Average Memory | 290MB | 368MB | 1,057MB | 4,834MB | 1,234MB |

Table 16: Synthesis speed for PeriodWave.

| Method | PeriodWave | PeriodWave | PeriodWave | PeriodWave-MB | PeriodWave-MB | PeriodWave-MB |
|---|---|---|---|---|---|---|
| Sampling Steps | 2 | 4 | 16 | 2 | 4 | 16 |
| Syn.Speed | 56.36× | 28.91 | 7.48× | 36.55× | 19.01× | 5.12× |
| Average Memory | 451MB | 453MB | 462MB | 424MB | 425MB | 432MB |

# E SYNTHESIS SPEED

We compared the synthesis speed and average memory usages on NVIDIA RTX A6000 GPU for each model. Table 15 indicated the synthesis speed for baseline models. HiFi-GAN shows the highest speed with a small memory usage. We reported the synthesis speed of our models according to sampling steps at Table 16. Although our model with sampling steps of 2 also has a better performance in objective evaluation than other models, our model required more time to generate higher quality samples with iterative generation. Period-wise batch inference could boost the inference speed by about 60%, but this also increases the average memory usage by about two times. For future work, we will reduce the inference speed by adopting adversarial learning with distillation methods.

Table 17: Grid Search for FreeU Hyperparameter.

| Methods | $\alpha$ | $\beta$ | M-STFT ($\downarrow$) | PESQ ($\uparrow$) | Periodicity ($\downarrow$) | V/UV F1 ($\uparrow$) | Pitch ($\downarrow$) | UTMOS ($\uparrow$) |
|---|---|---|---|---|---|---|---|---|
| PeriodWave | 1.00 | 1.00 | 1.2129 | 4.224 | 0.0762 | 0.9652 | 17.496 | 3.6495 |
| (16 steps) | 0.95 | 1.05 | 1.0975 | 4.253 | 0.0749 | 0.9660 | 17.503 | 3.7105 |
| | 0.94 | 1.06 | 1.0760 | 4.256 | 0.0752 | 0.9661 | 17.495 | 3.7163 |
| | 0.93 | 1.07 | 1.0590 | 4.255 | 0.0753 | 0.9658 | 17.450 | 3.7216 |
| | 0.92 | 1.08 | 1.0471 | 4.258 | 0.0757 | 0.9656 | 17.429 | 3.7263 |
| | 0.91 | 1.09 | 1.0394 | 4.254 | 0.0762 | 0.9655 | 17.417 | 3.7286 |
| | 0.90 | 1.10 | 1.0360 | 4.245 | 0.0765 | 0.9651 | 17.398 | 3.7307 |
| | 0.89 | 1.11 | 1.0364 | 4.240 | 0.0771 | 0.9647 | 17.363 | 3.7319 |
| | 0.88 | 1.12 | 1.0403 | 4.230 | 0.0777 | 0.9646 | 17.317 | 3.7340 |
| | 0.87 | 1.13 | 1.0472 | 4.213 | 0.0779 | 0.9643 | 17.184 | 3.7336 |
| | 0.86 | 1.14 | 1.0565 | 4.195 | 0.0784 | 0.9641 | 17.150 | 3.7330 |
| | 0.85 | 1.15 | 1.0682 | 4.173 | 0.0786 | 0.9640 | 17.156 | 3.7307 |
| | 0.80 | 1.20 | 1.1515 | 4.033 | 0.0812 | 0.9632 | 17.197 | 3.7139 |
| | 0.50 | 1.50 | 1.9347 | 2.572 | 0.1074 | 0.9457 | 25.241 | 3.3386 |
| | 0.95 | 1.00 | 1.0836 | 4.206 | 0.0764 | 0.9654 | 17.892 | 3.6262 |
| | 0.95 | 1.05 | 1.0975 | 4.253 | 0.0749 | 0.9660 | 17.503 | 3.7105 |
| | 0.95 | 1.10 | 1.1326 | 4.243 | 0.0762 | 0.9650 | 17.350 | 3.7456 |
| | 0.95 | 1.15 | 1.1819 | 4.164 | 0.0786 | 0.9637 | 17.097 | 3.7578 |
| | 0.95 | 1.20 | 1.2371 | 4.045 | 0.0847 | 0.9575 | 25.342 | 3.7477 |
| | 0.95 | 1.25 | 1.2936 | 3.872 | 0.0890 | 0.9555 | 25.429 | 3.7252 |
| | 0.90 | 1.00 | 1.0300 | 4.140 | 0.0753 | 0.9665 | 17.506 | 3.5755 |
| | 0.90 | 1.05 | 1.0124 | 4.234 | 0.0757 | 0.9657 | 17.522 | 3.6798 |
| | 0.90 | 1.10 | 1.0360 | 4.245 | 0.0765 | 0.9651 | 17.398 | 3.7307 |
| | 0.90 | 1.15 | 1.0892 | 4.183 | 0.0782 | 0.9642 | 17.079 | 3.7546 |
| | 0.90 | 1.20 | 1.1549 | 4.070 | 0.0821 | 0.9619 | 17.196 | 3.7530 |
| | 0.90 | 1.25 | 1.2222 | 3.906 | 0.0856 | 0.9596 | 19.787 | 3.7344 |
| | 0.90 | 1.30 | 1.2915 | 3.700 | 0.0921 | 0.9554 | 22.346 | 3.6934 |
| | 0.85 | 1.00 | 1.1256 | 3.930 | 0.0786 | 0.9638 | 20.564 | 3.4977 |
| | 0.85 | 1.05 | 1.0542 | 4.136 | 0.0772 | 0.9649 | 17.515 | 3.6255 |
| | 0.85 | 1.10 | 1.0382 | 4.196 | 0.0761 | 0.9657 | 17.452 | 3.6948 |
| | 0.85 | 1.15 | 1.0682 | 4.173 | 0.0786 | 0.9640 | 17.156 | 3.7307 |
| | 0.85 | 1.20 | 1.1191 | 4.079 | 0.0803 | 0.9636 | 17.173 | 3.7424 |
| | 0.85 | 1.25 | 1.1816 | 3.927 | 0.0850 | 0.9599 | 19.675 | 3.7320 |
| | 0.85 | 1.30 | 1.2502 | 3.728 | 0.0906 | 0.9569 | 19.905 | 3.7010 |
| | 0.80 | 1.00 | 1.3068 | 3.514 | 0.0815 | 0.9627 | 20.699 | 3.3817 |
| | 0.80 | 1.05 | 1.1911 | 3.885 | 0.0792 | 0.9643 | 17.507 | 3.5447 |
| | 0.80 | 1.10 | 1.1303 | 4.044 | 0.0784 | 0.9642 | 17.435 | 3.6382 |
| | 0.80 | 1.15 | 1.1267 | 4.086 | 0.0782 | 0.9643 | 17.086 | 3.6911 |
| | 0.80 | 1.20 | 1.1515 | 4.033 | 0.0812 | 0.9632 | 17.197 | 3.7139 |
| | 0.80 | 1.25 | 1.1954 | 3.911 | 0.0848 | 0.9599 | 19.626 | 3.7155 |
| | 0.80 | 1.30 | 1.2518 | 3.732 | 0.0898 | 0.9567 | 19.809 | 3.6956 |
| | 0.75 | 1.00 | 1.5238 | 3.000 | 0.0846 | 0.9578 | 26.287 | 3.2261 |
| | 0.75 | 1.05 | 1.3798 | 3.447 | 0.0813 | 0.9310 | 17.478 | 3.4308 |
| | 0.75 | 1.10 | 1.2835 | 3.738 | 0.0809 | 0.9632 | 17.435 | 3.5552 |
| | 0.75 | 1.15 | 1.2459 | 3.874 | 0.0810 | 0.9632 | 17.203 | 3.6325 |
| | 0.75 | 1.20 | 1.2414 | 3.902 | 0.0828 | 0.9626 | 17.329 | 3.6688 |
| | 0.75 | 1.25 | 1.2610 | 3.831 | 0.0851 | 0.9600 | 17.240 | 3.6840 |
| | 0.75 | 1.30 | 1.2993 | 3.690 | 0.9035 | 0.9562 | 19.798 | 3.6764 |
| | 0.70 | 1.00 | 1.7499 | 2.559 | 0.0862 | 0.9574 | 26.376 | 3.0309 |
| | 0.70 | 1.05 | 1.5887 | 2.949 | 0.0824 | 0.9622 | 17.481 | 3.2832 |
| | 0.70 | 1.10 | 1.4682 | 3.293 | 0.0834 | 0.9614 | 17.425 | 3.4427 |
| | 0.70 | 1.15 | 1.4024 | 3.524 | 0.0843 | 0.9616 | 17.271 | 3.5477 |
| | 0.70 | 1.20 | 1.3743 | 3.653 | 0.0852 | 0.9611 | 17.236 | 3.6053 |
| | 0.70 | 1.25 | 1.3711 | 3.662 | 0.0859 | 0.9609 | 17.262 | 3.6350 |
| | 0.70 | 1.30 | 1.3885 | 3.581 | 0.0912 | 0.9565 | 19.774 | 3.6395 |

## F FreeU

We search the hyperparameter of FreeU for a single-band model PeriodWave. We found that using balanced weights of backbone feature and skip feature could improve the reconstruction performance and perceptual quality. We utilized $\alpha$ of 0.9 and $\beta$ of 1.1 for our model. Additionally, we found that increasing the weight of backbone feature $\beta$ could further improve the perceptual quality but this would decrease the reproduction performance.

# G  TRAIN-INFERENCE MISMATCH PROBLEM

Current, two-stage Text-to-Speech (TTS) models consists of acoustic models and neural vocoder. Due to noisy Mel-spectrogram, these two-stage TTS models suffer from train-inference mismatch problem. Although one-step GAN-based neural vocoder could generate high-quality Mel-spectrogram, these models might generate the samples with a noisy sound due to train-inference mismatch problem.

To reduce this issue, HiFi-GAN (Kong et al., 2020) proposed the fine-tuning methods with the generated Mel-spectrogrm by teacher-forcing mode. (Lee et al., 2022a; Jang et al., 2021; Kaneko et al., 2022; Kim et al., 2020; Łańcucki, 2021) followed this fine-tuning method to improve the perceptual quality of two-stage TTS model.

Meanwhile, end-to-end TTS models (Kim et al., 2021; Lim et al., 2022) outperformed the performance compared to two-stage models in terms of audio quality. They have a limitation of model architecture restriction to align high-resolution waveform signal and text, and they require more training times. Additionally, recent end-to-end TTS models showed lower zero-shot TTS performance than recent two-stage TTS models including VoiceBox (Le et al., 2024), P-Flow (Kim et al., 2024), E2-TTS (Eskimez et al., 2024), ARDiT-TTS (Liu et al., 2024b), and DiTTo-TTS (Lee et al., 2024a).

Although recent TTS models have shown their powerful performance on zero-shot TTS, there are still train-inference mismatch problem which contains some noise on the generated Mel-spectrogram resulting noisy sound.

To address this issue, we shift our focus from one-step generation to the iterative sampling based waveform generation. Following diffusion-based neural vocoder (Koizumi et al., 2023; Jang et al., 2023; Huang et al., 2022b; Koizumi et al., 2022; Roman et al., 2023), waveform generation with iterative sampling could refine the waveform signal when the conditioning is flawed or imperfect. We also adopt the iterative sampling methods by optimizing flow matching objective to reduce the sampling steps. The results also show that our models have shown better performance even with small sampling steps. Furthermore, our model has shown the best performance on two-stage text-to-speech scenarios by iterative sampling.

## G.1  SINGLE SPEAKER TEXT-TO-SPEECH

We conduct two-stage TTS experiments to evaluate the robustness of the proposed models compared to previous GAN-based and diffusion-based models. We utilized the official implementation of Glow-TTS which is trained with the LJSpeech dataset. Table 18 demonstrated that our model has a higher performance on the two-stage TTS in terms of MOS and UTMOS. Although HiFi-GAN shows a lower performance in reconstruction metrics, we observed that HiFi-GAN shows a high perceptual perfor-

Table 18: Text-to-Speech Results. We utilized Glow-TTS trained with LJSpeech as TTS model.

| Methods | MOS (↑) | UTMOS (↑) |
|---|---|---|
| HiFi-GAN | 3.70±0.03 | 4.1114 |
| BigVGAN-base♥ | 3.71±0.03 | 4.0296 |
| BigVGAN♥ | 3.69±0.03 | 3.9570 |
| PriorGrad (50 steps) | 3.53±0.03 | 3.3807 |
| FreGrad (50 steps) | 3.51±0.03 | 2.8583 |
| PeriodWave (16 steps) | 3.72±0.03 | 4.2560 |
| PeriodWave + FreeU (16 steps) | **3.75±0.03** | **4.3110** |

mance in terms of UTMOS. BigVGAN-base (14M) has a higher performance than BigVGAN (112M). We see that BigVGAN could reconstruct the waveform signal from the generated Mel-spectrogram even with the error that might be in the generated Mel-spectrogram. Although our model has a higher reconstruction performance, our models could refine this phenomenon through iterative generative processes. Additionally, we found that the generated Mel-spectrogram contains a larger scale of energy compared to the ground-truth Mel-spectrogram, so we utilized $\tau$ of 0.333 for scaling $x_0$.

Table 19: Objective evaluation results with different sampling steps for each band.

| Method | steps | M-STFT (↓) | PESQ (↑) | Periodicity (↓) | V/UV F1 (↑) | UTMOS (↑) |
|---|---|---|---|---|---|---|
| PeriodWave-MB | [16,16,16,16] | 0.9729 | 4.262 | 0.0704 | 0.9678 | 3.6534 |
|  | [16,8,4,4] | 1.0473 | 4.259 | 0.0701 | 0.9680 | 3.6506 |
|  | [16,4,4,4] | 1.0580 | 4.257 | 0.0703 | 0.9678 | 3.6473 |
|  | [16,4,2,2] | 1.1148 | 4.255 | 0.0703 | 0.9678 | 3.6482 |
|  | [16,4,1,1] | 1.0883 | 4.241 | 0.0703 | 0.9677 | 3.6409 |
|  | [16,2,1,1] | 1.1033 | 4.224 | 0.0705 | 0.9677 | 3.6370 |
|  | [16,1,1,1] | 1.1133 | 4.200 | 0.0710 | 0.9677 | 3.6253 |
| PeriodWave-MB | [8,2,2,2] | 1.1428 | 4.239 | 0.0721 | 0.9670 | 3.6241 |
|  | [8,2,1,1] | 1.1152 | 4.225 | 0.0723 | 0.9669 | 3.6178 |
|  | [8,1,1,1] | 1.1255 | 4.193 | 0.0725 | 0.9670 | 3.6073 |
| PeriodWave-MB | [4,4,8,16] | 1.0609 | 4.235 | 0.0732 | 0.9671 | 3.5923 |
|  | [4,4,4,4] | 1.0825 | 4.232 | 0.0732 | 0.9670 | 3.5899 |

## G.2 ANALYSIS ON ADAPTIVE SAMPLING STEPS FOR MULTI-BAND MODELS

We proposed an adaptive sampling for multi-band models. We can efficiently reduce the sampling steps for high-frequency bands due to the hierarchical band modeling conditioned on the previously generated DWT components. Table 19 shows that it is important to model the first DWT components. After sampling the first band, we can significantly reduce the sampling steps for the remaining bands, maintaining the performance with only a small decrease. The results from the sampling steps of [4,4,8,16] demonstrated that it is important to model the first band for high-fidelity waveform generation and accurate high-frequency modeling could improve the M-STFT metrics.

## H BROADER IMPACT

**Practical Application** We first introduce a high-fidelity waveform generation model using flow matching. We demonstrated the out-of-distribution robustness of our model, and this means that the conventional neural vocoder can be replaced with our model. Furthermore, we train Codec-based PeriodWave for audio generation and speech language models. We see that our models can be utilized for text-to-speech, voice conversion, audio generation, and speech language models for high-quality waveform decoding.

**Social Negative Impact** Recently, speech AI technology has shown its practical applicability by synthesizing much more realistic audio. Unfortunately, this also increases the risk of the potential social negative impact including malicious use and ethical issues by deceiving people. It is important to discuss a countermeasure that can address these potential negative impacts such as fake audio detection, anti-spoofing techniques, and audio watermark generation.

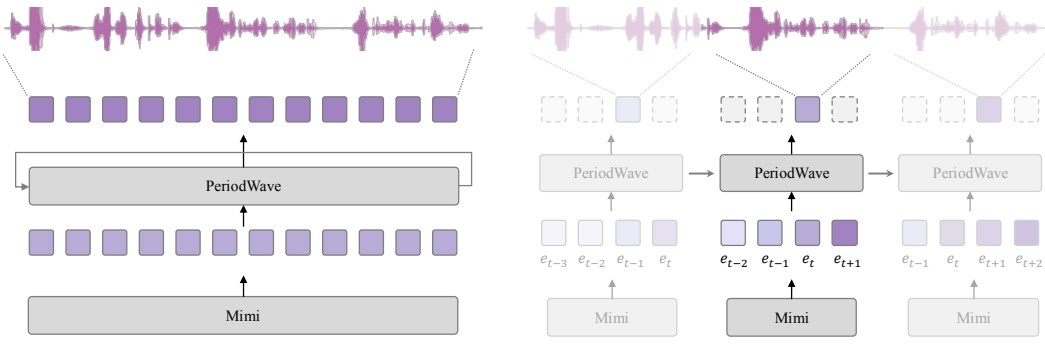

(a) Parallel generation          (b) Streaming generation

Figure 4: Neural Audio Codec Decoding. (a) The original PeriodWave is designed for parallel generation. Increasing the sampling steps can significantly improve the audio quality. (b) We propose streaming generation methods using a single token delayed streaming generation and 2-step sampling.

Table 20: Streaming audio generation using two steps based on different values of $N_p$ and $N_d$.

| $N_p$ | $N_d$ | CER (↓) | WER (↓) | M-STFT (↓) | PESQ (↑) | Periodicity (↓) | V/UV F1 (↑) | Pitch (↓) | UTMOS (↑) |
|---|---|---|---|---|---|---|---|---|---|
| 2 | 1 | 2.41 | 5.51 | 1.4386 | 2.056 | 0.1574 | 0.9121 | 50.098 | 3.6469 |
| 4 | 1 | 2.07 | 5.12 | 1.3364 | 2.174 | 0.1488 | 0.9182 | 46.104 | 3.7675 |
| 8 | 1 | 2.08 | 4.80 | 1.2889 | 2.211 | 0.1434 | 0.9243 | 47.302 | 3.8145 |
| 25 | 1 | 2.45 | 5.53 | 1.2716 | 2.233 | 0.1429 | 0.9228 | 40.634 | 3.8508 |

## I NEURAL AUDIO CODEC

Recently, neural audio codec models (Défossez et al., 2023; Yang et al., 2023a; Zhang et al., 2024; Liu et al., 2024a; Ji et al., 2024; Xin et al., 2024; Ye et al., 2024) have been investigated for practical applications including speech language models, audio generation, and TTS. Recently, Moshi (Défossez et al.) presented an efficient streaming speech language models and proposed neural audio codec, Mimi, that operates at 12.5 Hz. We train PeriodWave using tokens of $Q = 8$ quantizer from Mimi. We added an additional upsampling layer of Mel-Encoder, and we refer this module as the Token-Encoder. We utilize the same hyperparameter for the Mel-spectrogram version, excluding the segment length of 48,000. The model was trained on LibriTTS for 300 epochs (830k steps, 3 days).

**Fine-tuning with only adversarial training** Following (Lee et al., 2024b), we first fine-tuned the model with reconstruction loss and adversarial feedback, fixing the iteration steps to either 2 or 4 to accelerate the inference speed. However, we observed that the fine-tuned model using Mel-spectrogram reconstruction loss generated a noisy sound. We can discuss that Mimi compressed the waveform signal at 24,000 Hz into 12.5 Hz tokens, which causes a loss of frequency-related details in speech, resulting in over-smoothed frequency information with reconstruction loss. Moshi also indicated that removing the reconstruction loss can improve audio quality. Therefore, instead of Mel-reconstruction loss, we employed multi-scale STFTD with MPD and MS-SB-CQTD (Gu et al., 2024). With only adversarial training, we fine-tuned the model for 5 epochs (50k steps and 9 hours) with a batch size of 32 using four H100 GPUs.

**Parallel generation** We generated the samples using all sequence of tokens in parallel. We designed the model to enhance the audio quality by increasing the iteration steps. Table 10 indicates that PeriodWave significantly improves the performance compared to the original decoder of Mimi. Furthermore, we see that scaling up model size could further improve the audio quality in parallel generation for high-quality dialogue data collection using speech language models.

**Streaming generation** We present streaming generation methods for PeriodWave as illustrated in Figure 4. Although PeriodWave is trained with parallel generation, we found that PeriodWave can generate samples in a streaming manner. To achieve this, we utilize $N_p$ previous tokens for the past context, and $N_d$ delayed tokens for the future context. We adopt a single frame delayed streaming generation using $N_d = 1$ delayed token. We found that using at least one previous and one delayed token is essential for generating waveform signals in a streaming manner, as our model consists of non-causal convolutional layer. In our setup, we can use up to $N_p = 25$ previous tokens for real-time generation. We compared the performance based on different values of $N_p$ and $N_d$ in Table 20.

## J   BASELINE DETAILS

### J.1   LJSPEECH

We compared the model with the public-available models which are trained with LJSpeech dataset. LJSpeech is a single speaker dataset consisting of 13,100 high-quality audio samples with a sampling rate of 22,050 Hz. We followed the training and validation lists of HiFi-GAN[5].

**HiFi-GAN**   We first utilize the HiF-GAN, which is the most popular GAN-based neural vocoder. We use the official checkpoint of HiFi-GAN (V1)[6] which was trained for 2.5M steps. They utilize eight number of discriminators including three different scale of multi-scale discriminators and five different periods of multi-period discriminators.

**BigVGAN**   We utilize BigVGAN-base and BigVGAN which are a novel GAN-based neural vocoder. We utilize the official checkpoints[7] for sampling rate of 22,050 Hz which are trained with a large-scale dataset including LJSpeech, VCTK, and LibriTTS. We are not sure that they are trained with all LJSpeech dataset without splitting the training and validation dataset. These models are trained with 5M steps.

**PriorGrad**   We utilize PriorGrad which is the most popular diffusion-based neural vocoder. We use the official checkpoint of PriorGrad[8] which was trained for 3M steps. We used the same energy-based prior of this models and the default sampling steps of 50.

**FreGrad**   We utilize FreGrad Nguyen et al. (2024) which is the recent proposed diffusion-based neural vocoder. They utilize similar approach using discrete wavelet transform so we compare it with ours. We use the official checkpoint of FreGrad[9] which is trained for 1M steps.

### J.2   LIBRITTS

We compared the model with the public-available universal vocoder which are trained with LibriTTS dataset. LibriTTS dataset consists of 555 hours of 2,311 speakers with sampling rate of 24,000 Hz. We followed the training processes of BigVGAN including Mel-spectrogram transformation and inference settings. There are no diffusion-based models and any implementations which are trained with LibriTTS or other multi-speaker settings. In our preliminary study, diffusion-based models could not generate high-frequency information resulting in low quality audio generation.

**UnivNet**   We utilize the UnivNet-c32 which is a large model of UnivNet. UnivNet uses LVCNet which is an efficient generator structure for fast sampling. We use the public-available implementation of UnivNet[10] which is trained with LibriTTS train-clean-360 subset.

**Vocos**   We utilize Vocos wich is a fast time-frequency modeling-based neural vocoder with iSTFT. We utilize the official implementation of Vocos[11] which is trained with LibriTTS dataset for 1M steps. This model shows the fastest inference speed and even has a comparable performance with other baselines.

**BigVGAN**   We utilize BigVGAN-base and BigVGAN which are a novel GAN-based neural vocoder. We utilize the official checkpoints for sampling rate of 24,000 Hz which are trained with LibriTTS. These models are trained with 5M steps. Specifically, we also utilize the checkpoints which use a Snakebeta activation with log-scale parameterization which shows the best quality reported by the official implementation of BigVGAN.

---

[5] https://github.com/jik876/hifi-gan/tree/master/LJSpeech-1.1
[6] https://github.com/jik876/hifi-gan
[7] https://github.com/NVIDIA/BigVGAN
[8] https://github.com/microsoft/NeuralSpeech
[9] https://github.com/kaistmm/fregrad
[10] https://github.com/maum-ai/univnet
[11] https://github.com/gemelo-ai/vocos

## K    Evaluation Metrics

### K.1    Objective Evaluation

Following (Lee et al., 2023), we utilized four different metrics including multi-resolution STFT (M-STFT), perceptual evaluation of speech quality (PESQ), Periodicity error, and F1 score of voice/unvoice classification (V/UV F1). We additionally conduct UTMOS and Pitch distance.

**M-STFT**    we utilized an open-source implementation of multi-resolution STFT loss of Auraloss (Steinmetz & Reiss, 2020). The M-STFT loss was proposed in Parallel WaveGAN (Yamamoto et al., 2020), and we used this distance to measure the difference between the ground-truth and generated samples at the multiple resolution STFT domains.

**PESQ**    We utilized the wideband version of perceptual evaluation of speech quality[12]. We downsampled the audio by the sampling rate of 16,000 Hz to calculate PESQ.

**Periodicity and V/UV F1**    CarGAN (Morrison et al., 2022) stated the periodicity artifacts perceptually degrade the audio. We utilized a Periodicity RMSE to measure the periodicity error.[13] We also conducted the evaluation on Voice/Unvoice F1 score.

**UTMOS**    We utilize the open-source MOS prediction model, UTMOS[14] to evaluate the naturalness of generated samples. The UTMOS reported the consistency results of MOS for neutral English speech dataset.

### K.2    Subjective Evaluation

**MOS/SMOS**    We assessed the perceptual quality of synthesized speech using Mean Opinion Score (MOS). Specifically, the naturalness of the synthesized speech was measured with MOS, and its similarity to the ground-truth speech was evaluated using SMOS. We utilized crowdsourcing for this evaluation. The details are described in the following Appendix L.

---

[12]`https://github.com/ludlows/PESQ`
[13]`https://github.com/descriptinc/cargan`
[14]`https://github.com/tarepan/SpeechMOS`

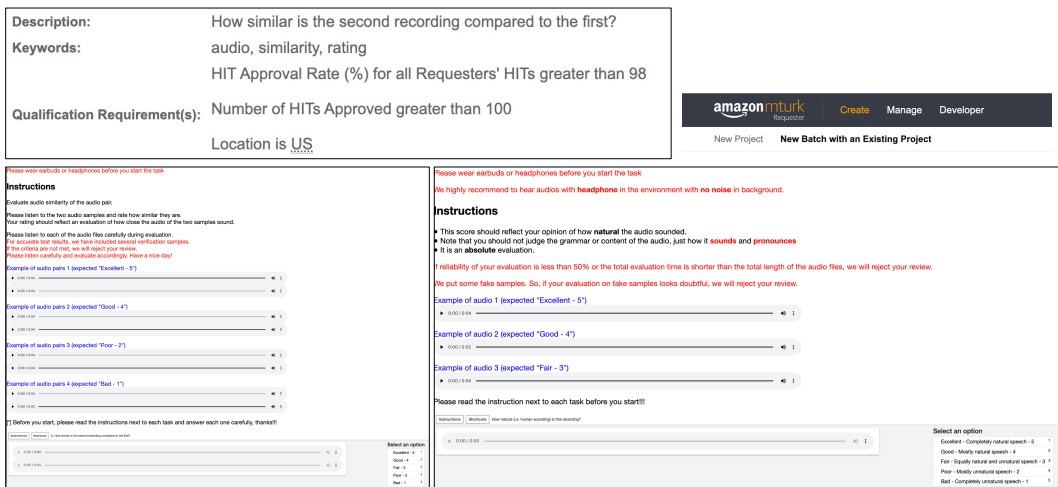

Figure 5: Detailed information on listeners restrictions and task completion interfaces.

## L    CROWDSOURCING DETAILS

We conducted MOS and SMOS evaluations using a 5-point scale to measure the naturalness and similarity of the synthesized speech. For this survey, we utilized Amazon Mechanical Turk[15] to assess the perceptual quality of each model. Specifically, 30 listeners evaluated 150 samples per model, rating them on a scale from 1 to 5. Given that the evaluation data is in English, we specifically targeted native English speakers residing in the United States. To ensure the reliability of the evaluators, we implemented strict eligibility criteria: only listeners with an approval rate of 98% or higher for their previous tasks on MTurk, and with at least 90 approved tasks (HiTs), were allowed to participate in this evaluation. To further enhance the quality of the evaluation, ground-truth samples were included as control measures. We excluded evaluations 1) from listeners who gave a score below 3 to the actual samples and 2) from those who spent less than half the duration of the audio sample on the overall evaluation from the final results. This procedure was intended to filter out inattentive listeners and ensure the integrity and accuracy of the evaluation data.

---

[15]https://www.mturk.com/

Table 21: Objective evaluation results from EnCodec tokens. We utilize speech dataset from the test samples provided in RFWave.

| Method | Params | Training | M-STFT (↓) | PESQ (↑) | Period. (↓) | V/UV (↑) | Pitch (↓) | UTMOS (↑) | SSL-MOS (↑) |
|---|---|---|---|---|---|---|---|---|---|
| Encodec | 15M | 300 epochs (8xA100) | 1.170 | 2.643 | 0.112 | 0.941 | 26.605 | 2.542 | 3.787 |
| Vocos | 7M | 2M steps | 1.074 | 3.051 | 0.086 | 0.957 | 20.491 | 3.100 | 4.130 |
| MBD | 411M | 4×2day (4×V100) | 1.612 | 2.645 | 0.108 | 0.946 | 26.720 | 3.300 | 4.091 |
| RFWave | 18M | 10 day (4×A100) | 1.280 | 3.020 | 0.078 | 0.957 | 18.126 | 2.988 | 4.169 |
| PeriodWave | 31M | 1M steps, 4 day (4×A100) | 0.929 | 3.644 | 0.070 | 0.968 | 19.538 | 3.581 | 4.352 |

Table 22: Band-wise Comparison of the reconstruction from EnCodec tokens. We utilize speech dataset from the test samples provided in RFWave.

| Method | Params | Training | Mel-L (↓) | Mel-M (↓) | Mel-H (↓) | Mel-A (↓) |
|---|---|---|---|---|---|---|
| Encodec | 15M | 300 epochs (8xA100) | 0.577 | 0.517 | 0.587 | 0.567 |
| Vocos | 7M | 2M steps | 0.449 | 0.406 | 0.451 | 0.441 |
| MBD | 411M | 4×2day (4×V100) | 0.837 | 0.994 | 1.107 | 0.921 |
| RFWave | 18M | 10 day (4×A100) | 0.603 | 0.674 | 0.775 | 0.651 |
| PeriodWave | 31M | 1M steps, 4 day (4×A100) | 0.386 | 0.365 | 0.413 | 0.387 |

Table 23: Objective evaluation results from EnCodec tokens. We utilize vocal dataset from the test samples provided in RFWave.

| Method | Params | Training | M-STFT (↓) | PESQ (↑) | Period. (↓) | V/UV (↑) | Pitch (↓) | Mel-L (↓) | Mel-M (↓) | Mel-H (↓) | Mel-A (↓) |
|---|---|---|---|---|---|---|---|---|---|---|---|
| Encodec | 15M | 300 epochs (8xA100) | 1.223 | 2.416 | 0.133 | 0.933 | 33.237 | 0.755 | 0.590 | 0.556 | 0.684 |
| Vocos | 7M | 2M steps | 1.131 | 2.849 | 0.098 | 0.954 | 24.504 | 0.546 | 0.457 | 0.454 | 0.511 |
| MBD | 411M | 4×2day (4×V100) | 1.554 | 2.894 | 0.096 | 0.957 | 22.252 | 0.891 | 1.024 | 1.095 | 0.957 |
| RFWave | 18M | 10 day (4×A100) | 1.345 | 2.878 | 0.079 | 0.962 | 18.849 | 0.760 | 0.690 | 0.701 | 0.735 |
| PeriodWave | 31M | 1M steps, 4 day (4×A100) | 0.921 | 3.681 | 0.067 | 0.967 | 17.531 | 0.448 | 0.391 | 0.415 | 0.431 |

Table 24: Objective evaluation results from EnCodec tokens. We utilize sound effect dataset from the test samples provided in RFWave.

| Method | Params | Training | M-STFT (↓) | PESQ (↑) | Mel-L (↓) | Mel-M (↓) | Mel-H (↓) | Mel-A (↓) |
|---|---|---|---|---|---|---|---|---|
| Encodec | 15M | 300 epochs (8xA100) | 1.182 | 2.409 | 0.528 | 0.466 | 0.456 | 0.502 |
| Vocos | 7M | 2M steps | 1.294 | 1.895 | 0.562 | 0.474 | 0.451 | 0.523 |
| MBD | 411M | 4×2day (4×V100) | 1.714 | 1.936 | 0.914 | 1.080 | 1.054 | 0.974 |
| RFWave | 18M | 10 day (4×A100) | 1.592 | 2.253 | 0.701 | 0.640 | 0.705 | 0.690 |
| PeriodWave | 31M | 1M steps, 4 day (4×A100) | 1.101 | 2.699 | 0.438 | 0.398 | 0.395 | 0.422 |

## M HIGH-FIDELITY WAVEFORM RECONSTRUCTION FROM THE TOKEN OF ENCODEC

We conducted additional experiments for waveform generation from the token of EncCodec, and we compared the model with EnCodec, Vocos, MBD and RFWave. Following MBD, we used the same Encodec settings to obtain the tokens, using a maximum bandwidth of 6.0 kbps (Q=8, 75Hz for 24,000 Hz waveform).

Following MBD, our model was trained and evaluated exclusively on the same dataset. Also, we utilized the official implementation and official checkpoints of MBD and RFWave, respectively. Specifically, we utilize the same dataset including Common Voice 7.0, DNSChallenge 4 for speech, MTG-Jamendo for music, and FSD50K and AudioSet for environmental sounds by resampling them using Sox resampling to 24,000 Hz. In summary, we utilized the same training dataset and our model was trained with speech, singing, and sound effects together for a fair comparison with MBD and RFWave.

We also incorporated the results from the universal test set provided by RFWave[16]. This testset consists of speech, vocal, and sound effect samples from several different datasets. Specifically, the reported results are based on the single-band PeriodWave model trained for 1M steps. For each model, we follow the recommended inference setup, MBD used 20 steps for each band diffusion,

---

[16]https://drive.google.com/file/d/1WjRRfD1yJSjEA3xfC8-635ugpLvnRK0f/view

we used 20 sampling steps for better results of RFWave with CFG of 2. We conducted additional naturalness evaluation by SSL-MOS. [17]

The results demonstrated that the single-band PeriodWave significantly outperforms all baselines including MBD and RFWave. With large-scale dataset, our model could increase the capacity to generalize various scenarios including speech, singing, and sound effects.

## M.1 BASELINES FOR ENCODEC TOKEN RECONSTRUCTION

**EnCodec** We utilized the official implementation of EnCodec Défossez et al. (2022), and the official checkpoint[18] to extract the tokens and reconstruct the waveform signal. We extract the tokens compressed by the bandwidth of 6.0 kbps (Q=8, 75Hz).

**Vocos** We use the official implementation and checkpoint of Vocos Siuzdak (2024) which was trained with the tokens of EnCodec.[19] Vocos is a powerful GAN-based baseline model. Although Vocos has a fast inference speed, Vocos was trained with multi-period discriminator and multi-resolution discriminator for adversarial feedback to optimize the model.

**MBD** We utilized the official implementation and checkpoint of multi-band diffusion (MBD) Roman et al. (2023).[20] Previous diffusion-based models only provide the low-frequency information, MBD adopted multi-band modeling for diffusion models to improve the high-frequency modeling.

**RFWave** We utilize the official implementation of RFWave Anonymous (2024) which has better performance than MBD. We found that RFWave was submitted to ICLR 2025[21], and this work also utilize a flow matching for waveform generation like our proposed methods. While our model adopt CFM to the waveform-level modeling with multi-period generation, RFWave leverage spectrogram-based flow matching for efficient iterative waveform generation.

---

[17]https://github.com/unilight/sheet
[18]https://github.com/facebookresearch/encodec
[19]https://github.com/gemelo-ai/vocos
[20]https://github.com/facebookresearch/audiocraft/blob/main/docs/MBD.md
[21]https://openreview.net/forum?id=gRmWtOnTLK

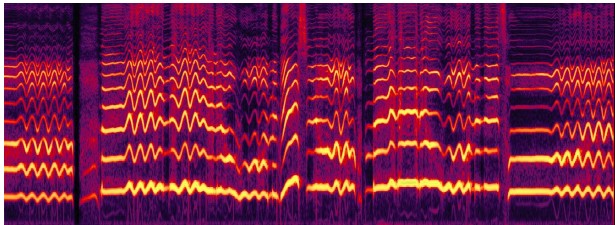
Figure 6: The spectrogram of the GT sample

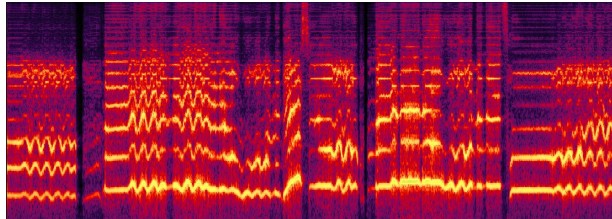
Figure 7: The spectrogram of the sample generated by EnCodec

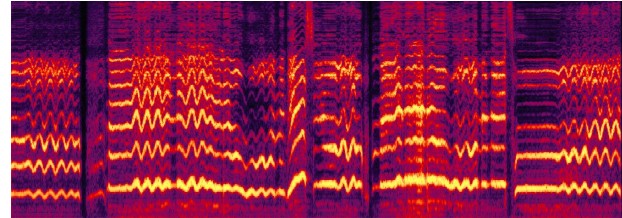
Figure 8: The spectrogram of the sample generated by Vocos

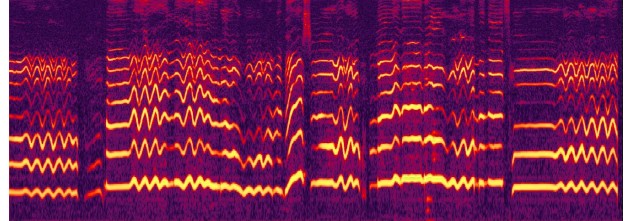
Figure 9: The spectrogram of the sample generated by MBD

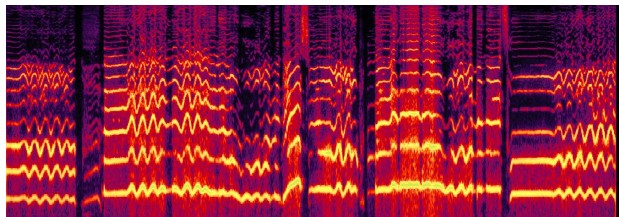
Figure 10: The spectrogram of the sample generated by RFWave

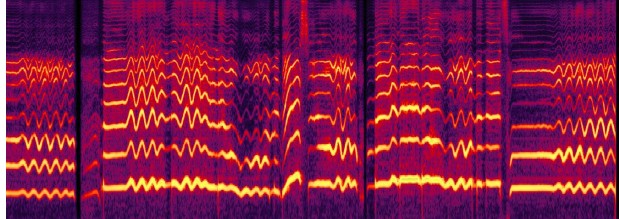
Figure 11: The spectrogram of the sample generated by PeriodWave

# N MULTI-BAND PERIODWAVE WITH FREEU

Table 25: Grid Search for FreeU Hyperparameter.

| Methods | $\alpha$ | $\beta$ | M-STFT ($\downarrow$) | PESQ ($\uparrow$) | Periodicity ($\downarrow$) | V/UV F1 ($\uparrow$) | Pitch ($\downarrow$) | UTMOS ($\uparrow$) |
|---|---|---|---|---|---|---|---|---|
| PeriodWave | 1.00 | 1.00 | 1.2129 | 4.224 | 0.0762 | 0.9652 | 17.496 | 3.6495 |
| (16 steps) | 0.95 | 1.05 | 1.0975 | 4.253 | 0.0749 | 0.9660 | 17.503 | 3.7105 |
|  | 0.90 | 1.10 | 1.0360 | 4.245 | 0.0765 | 0.9651 | 17.398 | 3.7307 |
|  | 0.85 | 1.15 | 1.0682 | 4.173 | 0.0786 | 0.9640 | 17.156 | 3.7307 |
| PeriodWave-MB | 1.00 | 1.00 | 0.9729 | 4.262 | 0.0704 | 0.9678 | 20.496 | 3.6534 |
| (16 steps) | 0.95 | 1.05 | 0.9590 | 4.291 | 0.0690 | 0.9689 | 20.235 | 3.7089 |
|  | 0.90 | 1.10 | 0.9671 | 4.283 | 0.0691 | 0.9687 | 20.138 | 3.7237 |
|  | 0.85 | 1.15 | 0.9876 | 4.224 | 0.0716 | 0.9673 | 20.016 | 3.7155 |

When we train the model with only a single CFM objective, the tendency of FreeU is almost same with other settings and dataset.

For validation set, we fixed the alpha and beta values in terms of reconstruction performance. However, we can decrease the alpha to remove the high-frequency noise, and increase the beta to improve the sharpness and energy of generated samples. In terms of generative models, controlling the alpha/beta could improve the diversity of generated samples.

Furthermore, we added the results of PeriodWave-MB with FreeU to demonstrate the consistent results. The results show a similar result of PeriodWave with FreeU.

