# OpenReview forum: "PeriodWave: Multi-Period Flow Matching for High-Fidelity Waveform Generation"
_ICLR.cc/2025/Conference — ICLR 2025 Poster_

### Official Review · Reviewer_C9NL · 2024-10-31

**Soundness:** 3
**Presentation:** 3
**Contribution:** 3
**Rating:** 6
**Confidence:** 3

**Summary:**

Propose PeriodWave, a novel universal waveform generator that can reflect different
implicit periodic information when estimating the vector fields.
Thoroughly analyze the limitation of high-frequency modeling, and address this by DWT and FreeU approach for high-frequency noise reduction.
Present PeriodWave which outperforms the one-step GAN models in conventional two-stage TTS tasks.
Based on SOTA neural audio codec Mimi, successfully demonstrate the effectiveness of
PeriodWave in neural audio codec decoding task both in parallel and streaming generation.

**Strengths:**

Thoroughly analyze the limitation of high-frequency modeling, and address this by DWT and FreeU approach for high-frequency noise reduction.
Present PeriodWave which outperforms the one-step GAN models in conventional two-stage TTS tasks.
Present thorough experimentation do demonstrate the effectiveness of the proposed method.

**Weaknesses:**

Although the models can generate the waveform with small sampling steps, Table E shows that the models have a slow synthesis speed compared to GAN-based neural vocoders.
Shows a lack of robustness in terms of high-frequency information because they only train the model by estimating the vector fields on the waveform resolution.

**Questions:**

Please add the limitations section to the main paper and remove it from the appendix.

---

> ### Author Response · Authors · 2024-11-18
>
> We sincerely appreciate your thorough and constructive feedback. We have carefully considered your suggestions and made changes to the best of our ability in an effort to alleviate your concerns.
>
> **W1. About the synthesis speed**
>
> We acknowledged that our models require more sampling steps compared to GAN-based Models. However, we hope to highlight the advantages of our model and the disadvantages of current GAN-based models as below
>
> **[1. Advantages of iterative sampling]** Our model could refine the generated audio through the iterative sampling processes. Unlike Diffusion or FM models, one-step GAN-based models may generate the noisy sound in a single step. Moreover, these phenomenon are more occurred in two-stage text-to-speech scenarios, resulting from the noisy generated Mel-spectrogram. We tried to clarify this point by conducting two-stage text-to-speech evaluation in Section 4.7 and 4.8.
>
> **[2. Performance]** While GAN-based Models show faster inference speed, the performance could not reach the performance of our models.
>
> **[3. Fast Training Speed]** Compared to GAN models, our model is trained with only a single FM objective. Unlike ours, current GAN models requires too many discriminator, resulting in slow training speed, hyper-parameter optimization, and instable training issues.
>
> We added the training time using the official implementation of BigVGAN and suggested hyperparameter. We used 4 A100 GPUs with 80GB GPU Memory. While our model was trained during 1M steps (2.4 days), BigVGAN was trained during 5M steps (40 days) and BigVSAN was trained during 10M steps (79 days). Furthermore, our model only requires 9GB of GPU memory for training, while BigVGAN requires 62GB and HiFi-GAN requires 26GB.
>
> Note that the performance of other GAN models with a small parameter significantly underperforms to ours and BigVGAN in OOD scenarios. Among GAN-based models, only BigVGAN demonstrates robustness in OOD scenarios. The official training time of HiFi-GAN is 13–14 days with V100 GPUs, as noted in its issues page. (https://github.com/jik876/hifi-gan/issues/7#issuecomment-719943371)
>
> In this regard, we do not consider HiFi-GAN and other GAN models to be on the same level as BigVGAN,  even though they require fewer training steps than BigVGAN. Furthermore, it is well known that GAN models require more training steps to optimize the model with various losses. Compared to BigVGAN and BigVSAN, our model has much better efficiency in training.
> We will clarify our statement for the training time of GAN models in the revised manuscript.
>
> **[Table R1]** The detailed training speed
>
> | Model | Objective | Time per 1M steps | Total Training Time | Segment  | Batch | Memory |
> | --- | --- | --- | --- | --- | --- | --- |
> | PeriodWave | Flow Matching | 2.4 days | 2.4 days (1M) | 32768 | 32 | 32GB |
> | PeriodWave | Flow Matching | 1 day     | -                     | 65536 | 4 | 9GB |
> | BigVGAN  | Recon. + GAN | 8 days | 40 days (5M)    | 65536 | 4 | 62GB |
> | BigVSAN  | Recon. + GAN | 7.9 days | 79 days (10M) | 65536 | 4 | 61GB |
> | HiFi-GAN | Recon. + GAN | 4 days  | 10 days (2.5M) | 65536 | 4 | 26GB |
>
> **[4. Relatively faster synthesis speed compared to diffusion models]** As Reviewer cFmR also summarized, our model achieves a good balance between quality and synthesis speed compared to conventional diffusion-based methods. We have added additional ablation study to compare the diffusion and CFM with the same model architectures as below:
>
> **[Table R2]** Comparison of CFM and Diffusion
> | Method | Steps | M-STFT (↓) | PESQ (↑) | Periodicity (↓) | V/UV F1 (↑) | Pitch (↓) | UTMOS (↑) |
> | --- | --- | --- | --- | --- | --- | --- | --- |
> | **PeriodWave w/ CFM** | 32 steps | 1.072 | 4.233 | 0.078 | 0.964 | 17.418 | 3.646 |
> | **PeriodWave w/ CFM** | 25 steps | 1.159 | 4.233 | 0.078 | 0.964 | 17.420 | 3.650 |
> | **PeriodWave w/ CFM** | 16 steps | 1.212 | 4.224 | 0.076 | 0.965 | 17.496 | 3.649 |
> | **PeriodWave w/ CFM** | 6 steps | 1.379 | 4.178 | 0.082 | 0.959 | 23.223 | 3.628 |
> | **PeriodWave w/ DDPM** | 50 steps | 1.159 | 4.151 | 0.084 | 0.961 | 23.046 | 3.377 |
> | **PeriodWave w/ DDPM** | 6 steps | 1.233 | 3.541 | 0.095 | 0.958 | 24.351 | 2.953 |
>
> The results show that the model with CFM has better performance even with a smaller sampling steps. We can also discuss that the model with CFM can converge faster than diffusion path and DDPM-based model might require more training steps to optimize the model.

---

> ### Author Response · Authors · 2024-11-18
>
> **W2. About the high-frequency information**
>
> As you mentioned, our model was only trained by estimating the vector filed on the waveform resolution. Hence, the generated samples might not reconstruct the details on spectral domain. However, it do not mean that the generated samples are low-quality. Our models have the best UTMOS and MOS score in objective and subjective naturalness evaluation. Additionally, diffusion-based models such as MBD and RFWave demonstrated that the diffusion and CFM has better perceptual quality while the objective metrics are lower.
>
> [Additional Comparison with MBD and RFWave] We have added additional comparison with multi-band diffusion models and RFWave to address your concern about high-frequency modeling. Following MBD, we used the same Encodec settings to obtain the tokens, using a maximum bandwidth of 6.0 kbps (Q=8, 75Hz for 24,000 Hz waveform). The results demonstrated that the single-band PeriodWave significantly outperforms MBD and RFWave in terms of M-STFT, PESQ, UTMOS metrics on encodec token reconstruction task.
>
> Due to limited time and resources, we could provide the single-band PeriodWave at this stage. We believe that PeriodWave-MB will have even better performance than the baselines.
>
> **[Table R3-1]** Objective evaluation results from EnCodec tokens. We compared the results using bandwidth of 6.0 kbps (Q=8, 75Hz) of EnCodec. MBD* is the results of MBD using loudness compressor provided in official implementation. We also evaluate MBD without loudness compressor, and we found that the original output of MBD contains white noise which are still not denoised by the diffusion processes.
>
> | Method | Params | Training | M-STFT (↓) | PESQ (↑) | Period. (↓) | V/UV (↑) | Pitch (↓) | UTMOS (↑) |
> | --- | --- | --- | --- | --- | --- | --- | --- | --- |
> | **Ground Truth** | - | - | - | - | - | - | - | 3.869 |
> | **MBD (20 steps)** | 411M | 4×2day (4×V100) | 1.569 | 2.737 | 0.123 | 0.940 | 29.065 | 3.797 |
> | **MBD (20 steps)*** | 411M | 4×2day (4×V100) | 2.025 | 2.577 | 0.123 | 0.939 | 28.863 | 3.724 |
> | **RFWave (10 steps, CFG2)** | 18M | 10 days (4×A100) | 1.418 | 3.119 | 0.093 | 0.953 | 24.420 | 3.529 |
> | **RFWave (20 steps, CFG2)** | 18M | 10 days (4×A100) | 1.346 | 3.098 | 0.093 | 0.953 | 24.156 | 3.521 |
> | **PeriodWave (16 steps)** | 31M | 500k steps, 2 days (4×A100) | 1.391 | 3.422 | 0.099 | 0.948 | 27.227 | 3.886 |
> | **PeriodWave+FreeU (16 steps)** | 31M | 500k steps, 2 days (4×A100) | 1.322 | 3.427 | 0.096 | 0.950 | 27.134 | 3.895 |
>
> **[Table R3-2]** Band-wise comparison of reconstruction results from EnCodec tokens. The table presents the band-wise Mel-spectrogram reconstruction performance of each model. Mel-L, Mel-M, Mel-H, and Mel-A represent the Mel-spectrogram distances for the 0–6 kHz, 6–12 kHz, 12–24 kHz, and 0–24 kHz frequency bands, respectively.
>
> | Method | Params | Training | Mel-L (↓) | Mel-M (↓) | Mel-H (↓) | Mel-A (↓) |
> | --- | --- | --- | --- | --- | --- | --- |
> | **MBD (20 steps)** | 411M | 4×2 days (4×V100) | 0.764 | 0.866 | 1.040 | 0.839 |
> | **MBD (20 steps)*** | 411M | 4×2 days (4×V100) | 0.779 | 0.859 | 1.186 | 0.875 |
> | **RFWave (10 steps, CFG2)** | 18M | 10 days (4×A100) | 0.614 | 0.815 | 1.018 | 0.733 |
> | **RFWave (20 steps, CFG2)** | 18M | 10 days (4×A100) | 0.580 | 0.701 | 0.898 | 0.667 |
> | **PeriodWave (16 steps)** | 31M | 500k steps, 2 days (4×A100) | 0.558 | 0.618 | **0.717** | **0.601** |
> | **PeriodWave+FreeU (16 steps)** | 31M | 500k steps, 2 days (4×A100) | **0.555** | **0.612** | 0.743 | 0.603 |
>
> Additionally, we compared the band-wise Mel distance (Mel-L:0-6 kHz, Mel-M: 6-12 kHz, Mel-H:12-24 kHz, and Mel-A:0-24 kHz). Our models have a lower Mel distance in terms of all bands including high-frequency band.
>
> ---
>
> **Q1. About the limitations section**
>
> Thanks for your suggestion. We will add the limitations section to the main paper to clarify the limitations of our models in the revised paper.

---

> > ### Author Response · Authors · 2024-11-25
> > **We would like to follow up on our rebuttal**
> >
> > Dear **Reviewer C9NL,**
> >
> > We sincerely thank you once again for your valuable time and effort in reviewing our manuscripts.
> >
> > We deeply appreciate your insightful comments and we have worked to address your concern through additional experiments and clarifications. We outline them as below:
> >
> > - **[A lack of robustness in terms of high-frequency information]** When we trained the model with a large-scale dataset using speech, vocals, and sound effects. Our model showed better robustness on the high-frequency domain compared to GAN-based models including EnCodec and Vocos. For a fair comparison, we follow the training setup and dataset of MBD and evaluated all models by the universal test set provided in RFWave. Furthermore, we evaluated the per-band Mel distance of all models using the universal test dataset including speech, vocal, sound effects, respectively (Pg. 28-29). We would like to share the results with the detailed spectrogram (Pg. 30) and demo samples (Demo page).
> > - **[Limitation Section]** We promise to move the limitation section to the main manuscripts at the final manuscripts.
> >
> > We hope these comments and improvements can be taken into account when deciding on the final score for paper. We remain open to further discussion and are happy to provide additional clarifications if needed.
> >
> > Thank you once again for your constructive feedback and the opportunity to improve our work.
> >
> > sincerely,
> >
> > The Authors.

---

### Official Review · Reviewer_nYdJ · 2024-11-04

**Soundness:** 3
**Presentation:** 3
**Contribution:** 3
**Rating:** 6
**Confidence:** 3

**Summary:**

The paper presents a nove waveform generation model that can capture periodic features of waveforms using a period-aware flow matching estimator and further extend to multi-band estimator.
Experimental results demonstrate that PeriodWave outperforms existing models in reconstruction tasks from mel and discrete tokens, as well as in TTS applications.

**Strengths:**

First time applying flow matching to waveform generation, and a method was designed to explicitly extract patterns with different periods to match the spectral characteristics of the waveforms.
Incorporates the latest effective waveform modeling tricks, significantly enhancing the performance of the vocoder and the extraction of periodic features.

**Weaknesses:**

1: There is no comparison of the performance differences between flow matching and diffusion under the same network and optimized configurations.

2: The main contribution of this paper lies in the introduction of periodic features; while the choice of periods [1, 2, 3, 5, 7] versus [1, 2, 4, 6, 8] can be understood, the ablation study for [1] may be unfair regarding network parameters (5 layers for [1, 2, 3, 5, 7] and only 1 layer for [1]). It would be more appropriate to compare with configurations like [1, 1, 1, 1, 1], [1, 1, 1, 3, 3], or [1, 1, 1, 3, 9]. Additionally, deriving the reason for selecting period by prime based solely on this comparison may not be sufficient，

3: The experiments emphasize that the network structure primarily optimizes high-frequency information modeling, but this does not adequately demonstrate that the method is specifically related to high-frequency information，It would be better to use band-wise comparison metrics.

**Questions:**

1: If α<1, wouldn't it make it harder for the model to capture high-frequency features?

2: The M-STFT metric is relatively low, a well-optimized model in the waveform space shouldn't show significant disadvantages in STFT metrics, is there any explanations?

3:A frequency of 22.05 kHz is not considered very high; it mainly contains harmonics, making it unclear whether the model's performance advantage stems from low-frequency modeling or high-frequency modeling.

---

> ### Author Response · Authors · 2024-11-18
>
> Thank you sincerely for your thoughtful comments and suggestions. We have carefully considered each of the points you raised and provided detailed responses below.
>
> **W1. About the comparison of the performance differences between flow matching and diffusion under the same networks and optimized configurations.**
>
> We have conducted additional comparisons between flow matching and diffusion models within the same multi-period generation structure, using the same optimized training configuration. For the diffusion, we choose PriorGrad-based DDPM objectives to utilize the same energy-based prior.  Furthermore, we adopted the optimized noise scheduling method and energy-based loss normalization proposed in PriorGrad, which are designed to improve DDPM-based Waveform generation tasks. We also observed that these techniques can improve the performance of DDPM-based models.
>
> The results show that the model with CFM has better performance even with a smaller sampling steps. We can discuss that the model with CFM can converge faster than diffusion path and DDPM-based model might require more training steps to optimize the model.
>
> **[Table R1]** Comparison of CFM and Diffusion
> | Method | Steps | M-STFT (↓) | PESQ (↑) | Periodicity (↓) | V/UV F1 (↑) | Pitch (↓) | UTMOS (↑) |
> | --- | --- | --- | --- | --- | --- | --- | --- |
> | **PeriodWave w/ CFM** | 32 steps | 1.072 | 4.233 | 0.078 | 0.964 | 17.418 | 3.646 |
> | **PeriodWave w/ CFM** | 25 steps | 1.159 | 4.233 | 0.078 | 0.964 | 17.420 | 3.650 |
> | **PeriodWave w/ CFM** | 16 steps | 1.212 | 4.224 | 0.076 | 0.965 | 17.496 | 3.649 |
> | **PeriodWave w/ CFM** | 6 steps | 1.379 | 4.178 | 0.082 | 0.959 | 23.223 | 3.628 |
> | **PeriodWave w/ DDPM** | 50 steps | 1.159 | 4.151 | 0.084 | 0.961 | 23.046 | 3.377 |
> | **PeriodWave w/ DDPM** | 6 steps | 1.233 | 3.541 | 0.095 | 0.958 | 24.351 | 2.953 |
>
> ---
>
> **W2. About the additional comparison with different periods [1, 1, 1, 1, 1]**
>
> Thanks for your feedback. We are pleased to have had the opportunity to discuss this matter in more detail! Based on your feedback, we have refined our statement regarding the introduction of periodic features. We have added the performance results for models trained with periods of [1, 1, 1, 1, 1], [1, 1, 1, 3, 3], and [1, 1, 1, 3, 9] as you mentioned.  The model with periods of [1, 1, 1, 1, 1] showed slightly better performance than the model with a periods of [1]. However, it significantly underperforms  compared to other models using different size of periods. The models with periods of [1, 1, 1, 3, 3] and [1, 1, 1, 3, 9] have better performance than [1, 1, 1, 1, 1], and the results demonstrate that using different periods could perform better.
>
> **[Table R2]** Comparison of models trained with different periodic features.
> | Method | Period | M-STFT (↓) | PESQ (↑) | Periodicity (↓) | V/UV F1 (↑) | UTMOS (↑) |
> | --- | --- | --- | --- | --- | --- | --- |
> | Ground Truth | - | - | - | - | - | 3.8626 |
> | PeriodWave | [1,2,3,5,7] | 1.1737 | 4.072 | 0.0806 | 0.9627 | 3.5544 |
> |  | [1] | 1.2588 | 3.795 | 0.0885 | 0.9572 | 3.4215 |
> |  | [1,1,1,1,1] | 1.1337 | 3.964 | 0.0888 | 0.9597 | 3.4728 |
> |  | [1,1,1,3,3] | 1.1234 | 4.011 | 0.0818 | 0.9643 | 3.4879 |
> |  | [1,1,1,3,9] | 1.2736 | 4.061 | 0.0830 | 0.9644 | 3.5057 |
> |  | [1,2,4,6,8] | 1.1481 | 4.075 | 0.0782 | 0.9647 | 3.5468 |
> |  | [1,2,4,8,16] | 1.1463 | 4.124 | 0.0787 | 0.9639 | 3.5408 |
> |  | [1,2,3,5,7,11,13,17] | 1.1617 | 4.125 | 0.0792 | 0.9610 | 3.5384 |
>
> ---
>
> **W3 & Q3. About the optimization of high-frequency information modeling**
>
> We have included an additional comparison with MBD and RFWave to address your concern regarding the modeling of high-frequency information modeling. Following MBD, we used the same Encodec settings to obtain the tokens, using a maximum bandwidth of 6.0 kbps (Q=8, 75Hz for 24,000 Hz waveform). The results demonstrated that the single-band PeriodWave significantly outperforms MBD and RFWave.
>
> **[Table R3]** Objective evaluation results from EnCodec tokens. MBD* results use the loudness compressor, while MBD without it shows white noise not removed by the diffusion process.
>
> | Method | Params | Training | M-STFT (↓) | PESQ (↑) | Period. (↓) | V/UV (↑) | Pitch (↓) | UTMOS (↑) |
> | --- | --- | --- | --- | --- | --- | --- | --- | --- |
> | **Ground Truth** | - | - | - | - | - | - | - | 3.869 |
> | **MBD (20 steps)** | 411M | 4×2day (4×V100) | 1.569 | 2.737 | 0.123 | 0.940 | 29.065 | 3.797 |
> | **MBD (20 steps)*** | 411M | 4×2day (4×V100) | 2.025 | 2.577 | 0.123 | 0.939 | 28.863 | 3.724 |
> | **RFWave (10 steps, CFG2)** | 18M | 10 days (4×A100) | 1.418 | 3.119 | 0.093 | 0.953 | 24.420 | 3.529 |
> | **RFWave (20 steps, CFG2)** | 18M | 10 days (4×A100) | 1.346 | 3.098 | 0.093 | 0.953 | 24.156 | 3.521 |
> | **PeriodWave (16 steps)** | 31M | 500k steps, 2 days (4×A100) | 1.391 | 3.422 | 0.099 | 0.948 | 27.227 | 3.886 |
> | **PeriodWave+FreeU (16 steps)** | 31M | 500k steps, 2 days (4×A100) | 1.322 | 3.427 | 0.096 | 0.950 | 27.134 | 3.895 |

---

> ### Author Response · Authors · 2024-11-18
>
> Due to limited time and resources, we could provide the single-band PeriodWave at this stage. We believe that PeriodWave-MB will have even better performance than the baselines. It is worth noting that our model has 13 times fewer parameters compared to MBD.
>
> Additionally, we compared the band-wise Mel distance across different frequency ranges: Mel-L (0–6 kHz), Mel-M (6–12 kHz), Mel-H (12–24 kHz), and Mel-A (0–24 kHz). Our models demonstrated a lower Mel distance across all bands.
>
> **[Table R4]** Comparison of Mel distance.
> | Method | Params | Training | Mel-L (↓) | Mel-M (↓) | Mel-H (↓) | Mel-A (↓) |
> | --- | --- | --- | --- | --- | --- | --- |
> | **MBD (20 steps)** | 411M | 4×2 days (4×V100) | 0.764 | 0.866 | 1.040 | 0.839 |
> | **MBD (20 steps)*** | 411M | 4×2 days (4×V100) | 0.779 | 0.859 | 1.186 | 0.875 |
> | **RFWave (10 steps, CFG2)** | 18M | 10 days (4×A100) | 0.614 | 0.815 | 1.018 | 0.733 |
> | **RFWave (20 steps, CFG2)** | 18M | 10 days (4×A100) | 0.580 | 0.701 | 0.898 | 0.667 |
> | **PeriodWave (16 steps)** | 31M | 500k steps, 2 days (4×A100) | 0.558 | 0.618 | **0.717** | **0.601** |
> | **PeriodWave+FreeU (16 steps)** | 31M | 500k steps, 2 days (4×A100) | **0.555** | **0.612** | 0.743 | 0.603 |
>
> ---
>
> **Q1. About alpha**
>
> We can discuss $\alpha$ in terms of reconstruction and generative models. First, the samples from previous diffusion models and our FM models at earlier time steps contain a lot of noise, and the energy of the high-frequency domain in the waveform is significantly lower than that of the low-frequency domain. However, the white noise we added has the same energy level across all frequencies of the waveform signal. In this regard, using $\alpha < 1$ could denoise the high-frequency noise at earlier steps, helping to restore the low-frequency information more robustly. During iterative sampling, we can also progressively restore high-frequency information. Thus, using $\alpha < 1$ could improve both reconstruction and perceptual quality. However, we found that this improves performance in the 0–12 kHz bandwidth, but slightly decreases performance in the 12–24 kHz bandwidth. We will add this discussion to the revised manuscript.
>
> However, Table 17 in the manuscript demonstrates that using a much lower $\alpha$ can further improve perceptual quality up to 0.85, and FreeU can enhance the diversity of generated samples in terms of generative models. It simply generates different samples, and this does not mean that the samples are of low quality.
>
> ---
>
> **Q2. About M-STFT metric**
>
> We believe that this result is expected, as the model was not regularized in the spectral domain. Recently, GAN-based models have incorporated Mel-spectrogram reconstruction loss and Multi-resolution Discriminators (such as the MRD in UnivNet, Vocos, and BigVGAN), which have been shown to enhance performance in the spectral domain.
> In fact, our model could be fine-tuned by incorporating spectral domain losses, as outlined **[Table R5]**. However, we hope to focus on the analysis of CFM for waveform modeling with a single FM objective in this paper.
>
> **[Table R5]** Comparison of M-STFT, PESQ, and other metrics across various models, with fine-tuning results for PeriodWave.
> | Method | Training Steps | Params (M) | M-STFT (↓) | PESQ (↑) | Period. (↓) | V/UV (↑) | Pitch (↓) | UTMOS (↑) |
> | --- | --- | --- | --- | --- | --- | --- | --- | --- |
> | **Ground Truth** | - | - | - | - | - | - | - | 4.3804 |
> | **HiFi-GAN (V1)** | 2.5M | 14.01 | 1.0341 | 3.646 | 0.1064 | 0.9584 | 26.839 | 4.2691 |
> | **BigVGAN-base** | 5.0M | 14.01 | 1.0046 | 3.868 | 0.1054 | 0.9597 | 25.142 | 4.1986 |
> | **BigVGAN** | 5.0M | 112.4 | 0.9369 | 4.210 | 0.0782 | 0.9713 | 19.019 | 4.2172 |
> | **BigVGAN-v2** | 3.0M | 112.4 | 0.8826 | 4.262 | 0.0663 | 0.9760 | 17.325 | 4.3110 |
> | **PriorGrad** | 3.0M | 2.61 | 1.2784 | 3.918 | 0.0879 | 0.9661 | 17.728 | 3.6282 |
> | **PeriodWave**  | 1.0M | 29.73 | 1.1464 | 4.288 | 0.0744 | 0.9704 | 15.042 | 4.3243 |
> | **PeriodWave+FreeU**  | 1.0M | 29.73 | 1.1132 | 4.293 | 0.0749 | 0.9701 | 15.753 | 4.3578 |
> | **PeriodWave + Finetuning** | 1K | 29.73 | 0.9253 | 4.349 | 0.0640 | 0.9766 | 14.943 | 4.3563 |
> | **PeriodWave + Finetuning** | 5K | 29.73 | 0.8969 | 4.344 | 0.0588 | 0.9783 | 13.662 | 4.3716 |
> | **PeriodWave + Finetuning** | 10K | 29.73 | 0.8880 | 4.387 | 0.0624 | 0.9769 | 14.186 | 4.3779 |
> | **PeriodWave + Finetuning** | 0.1M | 29.73 | **0.8352** | **4.418** | **0.0572** | **0.9786** | **13.473** | **4.3894** |

---

> > ### Author Response · Authors · 2024-11-25
> > **We would like to follow up on our rebuttal**
> >
> > Dear **Reviewer nYdJ,**
> >
> > We sincerely thank you once again for your valuable time and effort in reviewing our manuscripts.
> >
> > We deeply appreciate your insightful comments and we have worked to address your concern through additional experiments and clarifications. We outline them as below:
> >
> > - **[Comparison with CFM and Diffusion]** As you requested, we demonstrated the effectiveness of CFM for waveform modeling within the same experimental setup.
> > - **[Additional experiments for different periods]** We added the additional experiments for different periods including [1,1,1,1,1], [1,1,1,3,3], and [1,1,1,3,9] you requested.
> > - **[Additional analysis on high-frequency information]** In revised paper pg.28-30, we have added additional experiments using speech, vocal, and sound effects datasets to further demonstrate the effectiveness of each band modeling including low, middle, and high frequency. Follow MBD, we trained the model using the tokens of EnCodec, and evaluated all models using the universal test dataset including speech, vocal, sound effects, respectively. We would like to share the results with the detailed spectrogram (Pg. 30) and demo samples (Demo page).
> > - **[SOTA Performance in all evaluations]** We have demonstrated the superior performance of our model through extensive experiments. Compared to MBD (NeurIPS, 2023), BigVGAN (ICLR, 2023), Vocos (ICLR, 2024),  our model achieves state-of-the-art results. Furthermore, our model outperformed RFWave (Submitted in ICLR 2025), which received an average score of 6 (Submitted in ICLR2025). We hope this provided additional context when assessing the potential of our work.
> >
> > We hope these comments and improvements can be taken into account when deciding on the final score for paper. We remain open to further discussion and are happy to provide additional clarifications if needed.
> >
> > Thank you once again for your constructive feedback and the opportunity to improve our work.
> >
> > sincerely,
> >
> > The Authors.

---

### Official Review · Reviewer_cFmr · 2024-11-05

**Soundness:** 3
**Presentation:** 3
**Contribution:** 3
**Rating:** 6
**Confidence:** 2

**Summary:**

The paper proposed a flow-matching-based vocoder with improved high-frequency modeling and inference speed.

**Strengths:**

1. The paper proposed a practical flow-matching-based vocoder by significantly improving the high-frequency modeling issue with the existing flow-matching-based waveform generation method.
2. The proposed method shows decent performance compared to GAN-based methods.
3. The proposed method achieves a good balance between quality and synthesis speed compared to conventional diffusion-based methods.

**Weaknesses:**

1. It is unclear if the hyperparameters from grid searching during FreeU are generalizable to other settings and datasets. Does the alpha and beta need to be re-adjusted under different inference settings and datasets? It seems the FreeU grid search is only conducted on a single-band model of the proposed methods. Did you test it on the multi-band version of Periodwave?
2. Since high-frequency modeling is the core innovation of the paper, it is recommended to perform analyses focusing on the high-frequency modeling capabilities. For example, measuring the fidelity of the high-frequency bands of the audio signal and comparing it to the baselines. During the out-of-distribution music dataset experiment, only vocal, drum, and bass were emphasized, where the bass is typically the low-frequency part. It would be interesting to analyze the high-frequency components of the music by showing for example, per-band SNR accompanied with detailed spectrograms.

**Questions:**

See weaknesses.

---

> ### Author Response · Authors · 2024-11-18
>
> We sincerely appreciate the time and effort you have dedicated to providing us with such a detailed review. We are eager to thoughtfully address your questions and comments.
>
> **W1. About the FreeU hyperparameter**
>
> When we train the model with only a single CFM objective, the tendency of FreeU is almost same with other settings and dataset.
>
> For validation set, we fixed the alpha and beta values in terms of reconstruction performance. However, you can decrease the alpha to remove the high-frequency noise, and increase the beta to improve the sharpness and energy of generated samples. In terms of generative models, controlling the alpha/beta could improve the diversity of generated samples.
>
> We added the PeriodWave-MB with FreeU to address your concern. The results show a similar result of PeriodWave with FreeU.
>
> **[Table R1]** The performance results of PeriodWave and PeriodWave-MB integrated with FreeU.
>
> | Methods | α | β | M-STFT (↓) | PESQ (↑) | Periodicity (↓) | V/UV F1 (↑) | Pitch (↓) | UTMOS (↑) |
> | --- | --- | --- | --- | --- | --- | --- | --- | --- |
> | PeriodWave | 1.00 | 1.00 | 1.2129 | 4.224 | 0.0762 | 0.9652 | 17.496 | 3.6495 |
> | (16 steps) | 0.95 | 1.05 | 1.0975 | 4.253 | 0.0749 | 0.9660 | 17.503 | 3.7105 |
> |  | 0.90 | 1.10 | 1.0360 | 4.245 | 0.0765 | 0.9651 | 17.398 | 3.7307 |
> |  | 0.85 | 1.15 | 1.0682 | 4.173 | 0.0786 | 0.9640 | 17.156 | 3.7307 |
> | PeriodWave-MB | 1.00 | 1.00 | 0.9729 | 4.262 | 0.0704 | 0.9678 | 20.496 | 3.6534 |
> | (16 steps) | 0.95 | 1.05 | 0.9590 | 4.291 | 0.0690 | 0.9689 | 20.235 | 3.7089 |
> |  | 0.90 | 1.10 | 0.9671 | 4.283 | 0.0691 | 0.9687 | 20.138 | 3.7237 |
> |  | 0.85 | 1.15 | 0.9876 | 4.224 | 0.0716 | 0.9673 | 20.016 | 3.7155 |
> ---
>
> **W2. About the Band-wise Mel-spectrogram distance**
>
> We conducted additional comparison for waveform reconstruction from Encodec tokens to address your concern about high-frequency modeling. Following MBD, we used the same Encodec settings to obtain the tokens, using a maximum bandwidth of 6.0 kbps (Q=8, 75Hz for 24,000 Hz waveform). We compared Multi-band Diffusion models and RFWave for band-wise Mel-spectrogram Distance. Mel-L, Mel-M, Mel-H, and Mel-A denotes the Mel-spectrogram distance with bandwidths of 0-6, 6-12, 12-24, 0-24 kHz, respectively. Among diffusion and FM-based model, our model has the lowest error in all bands.
>
> **[Table R2]** Band-wise comparison of reconstruction results from EnCodec tokens. The table presents the band-wise Mel-spectrogram reconstruction performance of each model. MBD* is the results of MBD using loudness compressor provided in official implementation. We also evaluate MBD without loudness compressor, and we found that the original output of MBD contains white noise which are still not denoised by the diffusion processes.
>
> | Method | Params | Training | Mel-L (↓) | Mel-M (↓) | Mel-H (↓) | Mel-A (↓) |
> | --- | --- | --- | --- | --- | --- | --- |
> | **MBD (20 steps)** | 411M | 4×2 days (4×V100) | 0.764 | 0.866 | 1.040 | 0.839 |
> | **MBD (20 steps)*** | 411M | 4×2 days (4×V100) | 0.779 | 0.859 | 1.186 | 0.875 |
> | **RFWave (10 steps, CFG2)** | 18M | 10 days (4×A100) | 0.614 | 0.815 | 1.018 | 0.733 |
> | **RFWave (20 steps, CFG2)** | 18M | 10 days (4×A100) | 0.580 | 0.701 | 0.898 | 0.667 |
> | **PeriodWave (16 steps)** | 31M | 500k steps, 2 days (4×A100) | 0.558 | 0.618 | **0.717** | **0.601** |
> | **PeriodWave+FreeU (16 steps)** | 31M | 500k steps, 2 days (4×A100) | **0.555** | **0.612** | 0.743 | 0.603 |
> ---

---

> ### Author Response · Authors · 2024-11-25
> **We would like to follow up on our rebuttal**
>
> Dear **Reviewer cFmr,**
>
> We sincerely thank you once again for your valuable time and effort in reviewing our manuscripts.
>
> We deeply appreciate your insightful comments and we have worked to address your concern through additional experiments and clarifications. We outline them as below:
>
> - **[Multi-band PeriodWave with FreeU]** As you requested, we have added the results using PeriodWave-MB with FreeU.
> - **[Per-band Evaluation with extensive experiments using speech, vocal, and sound effects]** We have added additional comparison with MBD with the same training setting by EnCodec Token Reconstruction task. Follow MBD, we trained the model using the tokens of EnCodec, and evaluated the per-band Mel distance of all models using the universal test dataset including speech, vocal, sound effects, respectively (Pg. 28-29). We would like to share the results with the detailed spectrogram (Pg. 30) and demo samples (Demo page).
>
> We hope these comments and improvements can be taken into account when deciding on the final score for paper. We remain open to further discussion and are happy to provide additional clarifications if needed.
>
> Thank you once again for your constructive feedback and the opportunity to improve our work.
>
> sincerely,
>
> The Authors.

---

### Official Review · Reviewer_HbGY · 2024-11-06

**Soundness:** 3
**Presentation:** 4
**Contribution:** 3
**Rating:** 8
**Confidence:** 4

**Summary:**

The paper presents PeriodWave, the first flow-matching based vocoder. It relies on a carefully designed multi-period estimator. The authors notably propose to use discrete wavelet transforms and FreeU to better render high frequency information. The model outperforms one-step GANs in TTS tasks. The paper shows that the proposed method not only works as a spectrogram-based vocoder but also for generating from discrete tokens. Source code and checkpoints will be made available.

**Strengths:**

Extensive experimental work that demonstrates the robustness of the proposed method on both in-domain (mono- and multi-speaker TTS) and out-domain (Music) data. Ablations properly highlight the design choices.

**Weaknesses:**

I wish there was some comparison with MBD, which is probably the closest model architecture-wise.

**Questions:**

- Section 4.4: How is temperature exactly applied to the gaussian noise? Is this just about scaling the standard deviation?
- What is the motivation for not comparing with MDB at all in the paper? MBD is conceptually very similar (although only applied to discrete representations and using diffusion). Please motivate this choice somewhere in the paper.

---

> ### Author Response · Authors · 2024-11-18
>
> Thank you for your insightful comments and suggestions. We have thoroughly reviewed and responded to each of your questions and concerns below.
>
> **W1 & Q2. About the Comparison with MBD**
>
> For a fair comparison, we followed the experiments of BigVGAN. Furthermore, we hope to demonstrate the out-of-distribution (OOD) robustness of the model by training the model with only a speech dataset (LibriTTS). Specifically, we utilize the same train/dev filelists provided in BigVGAN official implementation. (https://github.com/NVIDIA/BigVGAN/tree/main/filelists/LibriTTS)
>
> However, we found it challenging to precisely reproduce  MBD's experimental settings. Their paper does not provide detailed experimental information such as filelists, and they employed pre-processing techniques like energy equalization. These aspects made it difficult to reproduce their results under the same conditions. Additionally, MBD utilized multiple datasets, which differs from our approach of using a speech dataset to evaluate OOD robustness.
>
> To address your concern, we added the comparison with MBD and RFWave. RFWave is a paper submitted in the same conference (ICLR2025) which also utilize multi-band settings and Rectified Flow. Following MBD, we used the same Encodec settings to obtain the tokens, using a maximum bandwidth of 6.0 kbps (Q=8, 75Hz for 24,000 Hz waveform). The results demonstrated that the single-band PeriodWave significantly outperforms MBD and RFWave. Due to limited time and resources, we could provide the single-band PeriodWave at this stage. We believe that PeriodWave-MB will have even better performance than the baselines. Additionally, our model has 13 times fewer parameters compared to MBD.
>
> **[Table R1-1]** Objective evaluation results from EnCodec tokens. We compared the results using bandwidth of 6.0 kbps (Q=8, 75Hz) of EnCodec. MBD* is the results of MBD using loudness compressor provided in official implementation. We also evaluate MBD without loudness compressor, and we found that the original output of MBD contains white noise which are still not denoised by the diffusion processes.
>
> | Method | Params | Training | M-STFT (↓) | PESQ (↑) | Period. (↓) | V/UV (↑) | Pitch (↓) | UTMOS (↑) |
> | --- | --- | --- | --- | --- | --- | --- | --- | --- |
> | **Ground Truth** | - | - | - | - | - | - | - | 3.869 |
> | **MBD (20 steps)** | 411M | 4×2day (4×V100) | 1.569 | 2.737 | 0.123 | 0.940 | 29.065 | 3.797 |
> | **MBD (20 steps)*** | 411M | 4×2day (4×V100) | 2.025 | 2.577 | 0.123 | 0.939 | 28.863 | 3.724 |
> | **RFWave (10 steps, CFG2)** | 18M | 10 days (4×A100) | 1.418 | 3.119 | 0.093 | 0.953 | 24.420 | 3.529 |
> | **RFWave (20 steps, CFG2)** | 18M | 10 days (4×A100) | 1.346 | 3.098 | 0.093 | 0.953 | 24.156 | 3.521 |
> | **PeriodWave (16 steps)** | 31M | 500k steps, 2 days (4×A100) | 1.391 | 3.422 | 0.099 | 0.948 | 27.227 | 3.886 |
> | **PeriodWave+FreeU (16 steps)** | 31M | 500k steps, 2 days (4×A100) | 1.322 | 3.427 | 0.096 | 0.950 | 27.134 | 3.895 |
>
> **[Table R1-2]** Objective evaluation results for EnCodec token reconstruction based on training steps.
>
> | Method | Training | M-STFT (↓) | PESQ (↑) | Period. (↓) | V/UV (↑) | Pitch (↓) | UTMOS (↑) |
> | --- | --- | --- | --- | --- | --- | --- | --- |
> | **PeriodWave (16 steps)** | 50k (4 hours) | 1.346 | 3.268 | 0.100 | 0.950 | 23.860 | 3.716 |
> | **PeriodWave (16 steps)** | 150k (12 hours) | 1.422 | 3.375 | 0.099 | 0.949 | 27.210 | 3.812 |
> | **PeriodWave (16 steps)** | 300k (1 day) | 1.440 | 3.417 | 0.101 | 0.948 | 26.790 | 3.875 |
> | **PeriodWave (16 steps)** | 500k (2 days) | 1.391 | 3.422 | 0.099 | 0.948 | 27.227 | 3.886 |
>
> **[Table R1-3]** Band-wise comparison of reconstruction results from EnCodec tokens. The table presents the band-wise Mel-spectrogram reconstruction performance of each model. Mel-L, Mel-M, Mel-H, and Mel-A represent the Mel-spectrogram distances for the 0–6 kHz, 6–12 kHz, 12–24 kHz, and 0–24 kHz frequency bands, respectively.
>
> | Method | Params | Training | Mel-L (↓) | Mel-M (↓) | Mel-H (↓) | Mel-A (↓) |
> | --- | --- | --- | --- | --- | --- | --- |
> | **MBD (20 steps)** | 411M | 4×2 days (4×V100) | 0.764 | 0.866 | 1.040 | 0.839 |
> | **MBD (20 steps)*** | 411M | 4×2 days (4×V100) | 0.779 | 0.859 | 1.186 | 0.875 |
> | **RFWave (10 steps, CFG2)** | 18M | 10 days (4×A100) | 0.614 | 0.815 | 1.018 | 0.733 |
> | **RFWave (20 steps, CFG2)** | 18M | 10 days (4×A100) | 0.580 | 0.701 | 0.898 | 0.667 |
> | **PeriodWave (16 steps)** | 31M | 500k steps, 2 days (4×A100) | 0.558 | 0.618 | **0.717** | **0.601** |
> | **PeriodWave+FreeU (16 steps)** | 31M | 500k steps, 2 days (4×A100) | **0.555** | **0.612** | 0.743 | 0.603 |
>
> ---
>
> **Q2. About the temperature**
> During inference, the noise is sampled from a multivariate normal distribution $\mathcal{N}(0, \Sigma)$. The sampled noise is then scaled by multiplying it with a temperature factor $\tau$, which adjusts the noise level during the generation process. We will clarify this point more clearly in the revised manuscript.

---

> ### Author Response · Authors · 2024-11-25
> **We would like to follow up on our rebuttal**
>
> Dear **Reviewer HbGY,**
>
> We sincerely thank you once again for your valuable time and effort in reviewing our manuscripts.
>
> We deeply appreciate your insightful comments and we have worked to address your concern through additional experiments and clarifications. We outline them as below:
>
> - **[Comparison with MBD]** As you requested, we have added additional comparison with MBD with the same training setting by EnCodec Token Reconstruction task. Follow MBD, we trained the model using the tokens of EnCodec, and evaluated all models using the universal test dataset including speech, vocal, sound effects, respectively (Pg. 28-29). We would like to share the results with the detailed spectrogram (Pg. 30) and demo samples (Demo page).
> - **[SOTA Performance in all evaluations]** We have demonstrated the superior performance of our model through extensive experiments. Compared to MBD (NeurIPS, 2023), BigVGAN (ICLR, 2023), Vocos (ICLR, 2024),  our model achieves state-of-the-art results. Furthermore, our model outperformed RFWave which is Submitted in ICLR 2025. We hope this provided additional context when assessing the potential of our work.
> - **[The analysis for better performance compared to MBD]** We conducted Multi-period ablation study using different periods at the original submission. The single-period model has a similar architecture design to MBD structure. Additionally, we demonstrated the effectiveness of CFM for waveform modeling by comparing the model trained with DDPM. Both CFM and Multi-period Architecture could improve the entire performance compared to MBD.
>
> We hope these comments and improvements can be taken into account when deciding on the final score for paper. We remain open to further discussion and are happy to provide additional clarifications if needed.
>
> Thank you once again for your constructive feedback and the opportunity to improve our work.
>
> sincerely,
>
> The Authors.

---

> > ### Comment · Reviewer_HbGY · 2024-11-26
> >
> > Dear Authors,
> >
> > Thanks for addressing my main concern. I think the paper definitely looks better with such comparison. I have updated my score accordingly.

---

> > > ### Author Response · Authors · 2024-11-27
> > >
> > > Dear Reviewer,
> > >
> > > Thank you for your thoughtful feedback and for updating your score. We greatly appreciate the time and effort you invested in reviewing our paper. Your comments have been invaluable in helping us improve the quality of our work.
> > >
> > >
> > > Best regards,
> > >
> > > Authors

---

### Official Review · Reviewer_TzrS · 2024-11-09

**Soundness:** 3
**Presentation:** 3
**Contribution:** 2
**Rating:** 6
**Confidence:** 4

**Summary:**

This paper studies the task of speech and audio waveform generation using the flow-matching modeling paradigm. The authors proposed a multi-period approach to generation artifacts in high-frequencies, conditioned on either mel-spectrogram and discrete representation extracted from neural codecs. The authors compared the proposed method to several GAN-based approaches and two diffusion-based methods. The authors empirically demonstrated the proposed method reaches superior performance to the evaluated baselines.

**Strengths:**

1. The proposed method reaches superior performance to the evaluated baselines.
2. The authors presented results using both mel-spectrogram and discrete representation extracted from neural codecs.
3. The authors provide extensive ablation and analysis of the proposed method.

**Weaknesses:**

1. Limited novelty. It seems the proposed method applies the multi-band (MB) approach (using wavelets instead of MB) over the flow-matching approach rather than diffusion.
2. Missing comparisons. As the proposed approach is heavily connected to the MBD approach [1], it is not clear why the authors did not compare to this method.
3. Missing details and inaccurate details (see below).


[1] San Roman, Robin, et al. "From discrete tokens to high-fidelity audio using multi-band diffusion." Advances in neural information processing systems 36 (2023): 1526-1538.

**Questions:**

I have a few questions and concerns about this submission:
1. As stated above, it seems the proposed method is heavily connected to the MBD work. What is the reason the authors did not compare their method to MBD? More generally, there are missing experiments to make the contribution of this work clear. At the moment, it is not clear what makes this method better than the prior work. Is it the flow-matching approach? Why not use v-diffusion instead? is it the usage of discrete wavelet transformation over MB? Etc.
**Specific experiments to add:** Can the authors provide: (1) a direct comparison of the proposed method MBD using the same setup? (2) ablation study on the usage of flow matching vs. diffusion? (3) comparison of discrete wavelet transform to the multi-band approach used in MBD?
2. There are missing details regarding the evaluation methods. Can the authors define the "Pitch" metric mean? How did the authors extract F0 information? What metric did the authors use to estimate F0 quality? Similarly for the other metrics reported in the paper.
3. Under the training paragraph, the authors mention that training (using 2 A100s) the proposed approach is faster as it requires 3 days, while it takes more than three weeks to train the GAN-based approaches. It seems quite a lot for the GAN-based methods. It makes sense training the proposed method would be faster, however, it seems quite a big difference. Can the authors provide more information on how they trained the baseline methods? More details about the training configurations for both the proposed method and the evaluated baselines? (i.e., batch sizes, learning rates, etc.). Alternatively, in case such training times were reported in prior work, a citation would be fine.

---

> ### Author Response · Authors · 2024-11-18
>
> Thank you for your valuable comments and suggestions. We have carefully addressed each of your questions and concerns below. Please feel free to let us know if there are any aspects that require further clarification.
>
>
> **W1. About the multi-band approach**
>
> We hope to address your concern about the novelty. Although the multi-band approach can contribute to the performance of our framework, it is one of the additional options and not the sole or primary contribution. The main contributions of our work include: 1) The Multi-Period Generator structure, 2) An optimized CFM training method for waveform generation, and 3) The first attempt to analyze high-frequency understanding using FreeU and Multi-Band. Specifically, we demonstrate the single-band models integrated with FreeU outperform the baselines. Additionally, multi-band models can leverage FreeU to further enhance performance by suppressing high-frequency noise, as shown below:
>
> **[Table R1]** The performance results of PeriodWave and PeriodWave-MB integrated with FreeU.
> | Methods | α | β | M-STFT (↓) | PESQ (↑) | Periodicity (↓) | V/UV F1 (↑) | Pitch (↓) | UTMOS (↑) |
> | --- | --- | --- | --- | --- | --- | --- | --- | --- |
> | PeriodWave | 1.00 | 1.00 | 1.2129 | 4.224 | 0.0762 | 0.9652 | 17.496 | 3.6495 |
> | (16 steps) | 0.95 | 1.05 | 1.0975 | 4.253 | 0.0749 | 0.9660 | 17.503 | 3.7105 |
> |  | 0.90 | 1.10 | 1.0360 | 4.245 | 0.0765 | 0.9651 | 17.398 | 3.7307 |
> |  | 0.85 | 1.15 | 1.0682 | 4.173 | 0.0786 | 0.9640 | 17.156 | 3.7307 |
> | PeriodWave-MB | 1.00 | 1.00 | 0.9729 | 4.262 | 0.0704 | 0.9678 | 20.496 | 3.6534 |
> | (16 steps) | 0.95 | 1.05 | 0.9590 | 4.291 | 0.0690 | 0.9689 | 20.235 | 3.7089 |
> |  | 0.90 | 1.10 | 0.9671 | 4.283 | 0.0691 | 0.9687 | 20.138 | 3.7237 |
> |  | 0.85 | 1.15 | 0.9876 | 4.224 | 0.0716 | 0.9673 | 20.016 | 3.7155 |
>
> ---
>
> **W2 & Q1. About the MBD comparison with the same setup**
>
> For a fair comparison, we followed the experiments of BigVGAN. Furthermore, we hope to demonstrate the out-of-distribution (OOD) Robustness of the model by training the model with only a speech dataset (LibriTTS). Specifically, we utilize the same train/dev filelists provided in BigVGAN official implementation. (https://github.com/NVIDIA/BigVGAN/tree/main/filelists/LibriTTS)
>
> However, we found it challenging to precisely reproduce  MBD's experimental settings. Their paper does not provide detailed experimental information such as filelists, and they employed pre-processing techniques like energy equalization. These aspects made it difficult to reproduce their results under the same conditions. Additionally, MBD utilized multiple datasets, which differs from our approach of using a speech dataset to evaluate OOD robustness.
>
> To address your concern, we added the comparison with MBD and RFWave. RFWave is a paper submitted in the same conference (ICLR2025) which also utilize multi-band settings and Rectified Flow. Following MBD, we used the same Encodec settings to obtain the tokens, using a maximum bandwidth of 6.0 kbps (Q=8, 75Hz for 24,000 Hz waveform). The results demonstrated that the single-band PeriodWave significantly outperforms MBD and RFWave. Due to limited time and resources, we could provide the single-band PeriodWave at this stage. We believe that PeriodWave-MB will have even better performance than the baselines. Additionally, our model has 13 times fewer parameters compared to MBD.

---

> ### Author Response · Authors · 2024-11-18
>
> **[Table R2-1]** Objective evaluation results from EnCodec tokens. We compared the results using bandwidth of 6.0 kbps (Q=8, 75Hz) of EnCodec. MBD* is the results of MBD using loudness compressor provided in official implementation. We also evaluate MBD without loudness compressor, and we found that the original output of MBD contains white noise which are still not denoised by the diffusion processes.
>
> | Method | Params | Training | M-STFT (↓) | PESQ (↑) | Period. (↓) | V/UV (↑) | Pitch (↓) | UTMOS (↑) |
> | --- | --- | --- | --- | --- | --- | --- | --- | --- |
> | **Ground Truth** | - | - | - | - | - | - | - | 3.869 |
> | **MBD (20 steps)** | 411M | 4×2day (4×V100) | 1.569 | 2.737 | 0.123 | 0.940 | 29.065 | 3.797 |
> | **MBD (20 steps)*** | 411M | 4×2day (4×V100) | 2.025 | 2.577 | 0.123 | 0.939 | 28.863 | 3.724 |
> | **RFWave (10 steps, CFG2)** | 18M | 10 days (4×A100) | 1.418 | 3.119 | 0.093 | 0.953 | 24.420 | 3.529 |
> | **RFWave (20 steps, CFG2)** | 18M | 10 days (4×A100) | 1.346 | 3.098 | 0.093 | 0.953 | 24.156 | 3.521 |
> | **PeriodWave (16 steps)** | 31M | 500k steps, 2 days (4×A100) | 1.391 | 3.422 | 0.099 | 0.948 | 27.227 | 3.886 |
> | **PeriodWave+FreeU (16 steps)** | 31M | 500k steps, 2 days (4×A100) | 1.322 | 3.427 | 0.096 | 0.950 | 27.134 | 3.895 |
>
> **[Table R2-2]** Objective evaluation results for EnCodec token reconstruction based on training steps.
>
> | Method | Training | M-STFT (↓) | PESQ (↑) | Period. (↓) | V/UV (↑) | Pitch (↓) | UTMOS (↑) |
> | --- | --- | --- | --- | --- | --- | --- | --- |
> | **PeriodWave (16 steps)** | 50k (4 hours) | 1.346 | 3.268 | 0.100 | 0.950 | 23.860 | 3.716 |
> | **PeriodWave (16 steps)** | 150k (12 hours) | 1.422 | 3.375 | 0.099 | 0.949 | 27.210 | 3.812 |
> | **PeriodWave (16 steps)** | 300k (1 day) | 1.440 | 3.417 | 0.101 | 0.948 | 26.790 | 3.875 |
> | **PeriodWave (16 steps)** | 500k (2 days) | 1.391 | 3.422 | 0.099 | 0.948 | 27.227 | 3.886 |
>
> **[Table R2-3]** Band-wise comparison of reconstruction results from EnCodec tokens. The table presents the band-wise Mel-spectrogram reconstruction performance of each model. Mel-L, Mel-M, Mel-H, and Mel-A represent the Mel-spectrogram distances for the 0–6 kHz, 6–12 kHz, 12–24 kHz, and 0–24 kHz frequency bands, respectively.
>
> | Method | Params | Training | Mel-L (↓) | Mel-M (↓) | Mel-H (↓) | Mel-A (↓) |
> | --- | --- | --- | --- | --- | --- | --- |
> | **MBD (20 steps)** | 411M | 4×2 days (4×V100) | 0.764 | 0.866 | 1.040 | 0.839 |
> | **MBD (20 steps)*** | 411M | 4×2 days (4×V100) | 0.779 | 0.859 | 1.186 | 0.875 |
> | **RFWave (10 steps, CFG2)** | 18M | 10 days (4×A100) | 0.614 | 0.815 | 1.018 | 0.733 |
> | **RFWave (20 steps, CFG2)** | 18M | 10 days (4×A100) | 0.580 | 0.701 | 0.898 | 0.667 |
> | **PeriodWave (16 steps)** | 31M | 500k steps, 2 days (4×A100) | 0.558 | 0.618 | **0.717** | **0.601** |
> | **PeriodWave+FreeU (16 steps)** | 31M | 500k steps, 2 days (4×A100) | **0.555** | **0.612** | 0.743 | 0.603 |
>
> ---
> **W3 & Q1. Ablation study on the usage of flow matching vs. diffusion**
>
> Thanks for your suggestion. We have conducted additional comparisons between flow matching and diffusion models within the same multi-period generation structure, using the same optimized training configuration. For the diffusion, we choose PriorGrad-based DDPM objectives to utilize the same energy-based prior.
>
> Furthermore, we adopted the optimized noise scheduling method and energy-based loss normalization proposed in PriorGrad, which are designed to improve DDPM-based Waveform generation tasks. We also observed that these techniques can improve the performance of DDPM-based models.
>
> The results show that the model with CFM has better performance even with a smaller sampling steps. We can discuss that the model with CFM can converge faster than diffusion path and DDPM-based model might require more training steps to optimize the model.
>
> **[Table R3]** Comparison of CFM and Diffusion
> | Method | Steps | M-STFT (↓) | PESQ (↑) | Periodicity (↓) | V/UV F1 (↑) | Pitch (↓) | UTMOS (↑) |
> | --- | --- | --- | --- | --- | --- | --- | --- |
> | **PeriodWave w/ CFM** | 32 steps | 1.072 | 4.233 | 0.078 | 0.964 | 17.418 | 3.646 |
> | **PeriodWave w/ CFM** | 25 steps | 1.159 | 4.233 | 0.078 | 0.964 | 17.420 | 3.650 |
> | **PeriodWave w/ CFM** | 16 steps | 1.212 | 4.224 | 0.076 | 0.965 | 17.496 | 3.649 |
> | **PeriodWave w/ CFM** | 6 steps | 1.379 | 4.178 | 0.082 | 0.959 | 23.223 | 3.628 |
> | **PeriodWave w/ DDPM** | 50 steps | 1.159 | 4.151 | 0.084 | 0.961 | 23.046 | 3.377 |
> | **PeriodWave w/ DDPM** | 6 steps | 1.233 | 3.541 | 0.095 | 0.958 | 24.351 | 2.953 |

---

> ### Author Response · Authors · 2024-11-18
>
> **Q1. Comparison of DWT to the multi-band approach used in MBD**
> In our preliminary experiments, we could not reproduce the quality of multi-band approach used in MBD. When we followed the provided setting without energy equalization for a fair comparison, the generated samples contained a lot of white noise and specific band frequency noise. We also found that others have encountered the same problems:(https://github.com/facebookresearch/audiocraft/issues/430)
>
> Subsequently, we compared PQMF-based sub-band modeling and DWT. We found that using four channels with PQMF resulted in information loss during the inverse process. On the other hand, DWT is invertible, leading to a higher PESQ score of 4.50 compared to 4.07 for PQMF. In this regard, we select the DWT-based lossless band splitting method for our multi-band settings.
>
> ---
> **Q2. About the pitch metric**
>
> Thanks for your feedback. We missed the details of Pitch metric. We utilize the same Pitch metrics provided in CARGAN [ICLR, 2022], which include Periodicity, V/UV F1, and Pitch Metrics using pitch prediction model, CREPE. We will add the details of Pitch metric to the revised manuscript.
>
> ---
> **Q3. The detailed training speed**
>
> We add the training time using the official implementation of BigVGAN and suggested hyperparameter. We used 4 A100 GPUs with 80GB GPU Memory. While our model was trained during 1M steps (2.4 days), BigVGAN was trained during 5M steps (40 days) and BigVSAN was trained during 10M steps (79 days). Furthermore, our model only requires 9GB of GPU memory for training, while BigVGAN requires 62GB and HiFi-GAN requires 26GB.
>
> Note that the performance of other GAN models with a small parameter significantly underperforms to ours and BigVGAN in OOD scenarios. Among GAN-based models, only BigVGAN demonstrates robustness in OOD scenarios. The official training time of HiFi-GAN is 13–14 days with V100 GPUs, as noted in its issues page. (https://github.com/jik876/hifi-gan/issues/7#issuecomment-719943371)
>
> In this regard, we do not consider HiFi-GAN and other GAN models to be on the same level as BigVGAN,  even though they require fewer training steps than BigVGAN. Furthermore, it is well known that GAN models require more training steps to optimize the model with various losses. Compared to BigVGAN and BigVSAN, our model has much better efficiency in training.
> We will clarify our statement for the training time of GAN models in the revised manuscript.
>
> **[Table R4]** The detailed training speed
>
> | Model | Objective | Time per 1M steps | Total Training Time | Segment  | Batch | Memory |
> | --- | --- | --- | --- | --- | --- | --- |
> | PeriodWave | Flow Matching | 2.4 days | 2.4 days (1M) | 32768 | 32 | 32GB |
> | PeriodWave | Flow Matching | 1 day     | -                     | 65536 | 4 | 9GB |
> | BigVGAN  | Recon. + GAN | 8 days | 40 days (5M)    | 65536 | 4 | 62GB |
> | BigVSAN  | Recon. + GAN | 7.9 days | 79 days (10M) | 65536 | 4 | 61GB |
> | HiFi-GAN | Recon. + GAN | 4 days  | 10 days (2.5M) | 65536 | 4 | 26GB |

---

> > ### Comment · Reviewer_TzrS · 2024-11-23
> > **Official Comment by Reviewer TzrS**
> >
> > I would like to thank the authors for providing additional details, experiments, and clarifications. This is highly appreciated.
> >
> > **Regarding the comparison to MBD:** I would like to thank the authors for providing the additional results and comparison, however, I'm afraid this is not a fair comparison, as the models were trained on different datasets. Moreover, it is not clear what datasets are used for evaluation. I believe the authors could have trained MBD and compared to their method under the same setup, the code for training MBD is publicly available: https://github.com/facebookresearch/audiocraft/blob/main/docs/MBD.md#training. Can the authors provide more details about the data used to evaluate their method against MBD and RFWave? Does it contain speech data only? speech+music? etc.
> > Additionally, the authors use a single period only for this comparison. In that case, can the authors provide details/intuition on what makes their method better than the MBD? is it the CFM objective vs. the diffusion approach used in MBD? Training data? Specific model architecture? It would be much easier to understand the contribution of the proposed work.
> >
> > **Regarding training speed:** Thanks for providing the exact training time details. It is not clear to me why the proposed method trains faster. Did the authors introduce specific components to make it train faster? Is it just the CFM objective compared to the GAN-based approach that makes the model converge faster? Is it the multi-period approach? The authors should justify it with experiments.

---

> ### Author Response · Authors · 2024-11-23
> **Thanks for your Reply**
>
> Thanks for your response!
>
> **[About the details of evaluation for the comparison to MBD and RFWave]**
>
> Thank you for your feedback. We would like to clarify that the comparison with MBD and RFWave was conducted with fairness and consistency. As outlined in the dataset section of MBD and RFWave, our model was trained and evaluated exclusively on the same dataset. Also, we utilized the official implementation and official checkpoints of MBD and RFWave, respectively.
>
> Specifically, we utilize the same dataset including Common Voice 7.0, DNSChallenge 4 for speech, MTG-Jamendo for music, and FSD50K and AudioSet for environmental sounds by resampling them using Sox resampling to 24,000 Hz.
>
> In summary, we utilized the **same training dataset** and our model was trained with speech, singing, and sound effects together for **a fair comparison with MBD and RFWave**.
>
> **[Additional Experiments of Speech, Vocal, and Sound Effects]**
>
> Previously, we provided the results from the LibriTTS benchmark subset of BigVGAN.
>
> We also incorporated the results from the universal test set provided by RFWave (https://drive.google.com/file/d/1WjRRfD1yJSjEA3xfC8-635ugpLvnRK0f/view). Specifically, we have further trained our model using a large-scale dataset for 1M steps. The reported results are based on the single-band PeriodWave model trained for 1M steps. MBD (20 steps), RFWave (20 steps, CFG2), PeriodWave (16 steps, FreeU). We conducted additional naturalness evaluation by SSL-MOS following https://github.com/unilight/sheet
>
> **<Table R5-1: Objective evaluation results from EnCodec tokens. We utilize speech dataset from the test samples provided in RFWave.>**
>
> | **Method** | **Params** | **Training**| **M-STFT (↓)** | **PESQ (↑)** | **Period. (↓)** | **V/UV (↑)** | **Pitch (↓)** | **UTMOS (↑)** | **SSL-MOS (↑)** |
> |-|-|-|-|-|-|-|-|-|-|
> | **MBD** | 411M | 4×2day (4×V100) |1.612| 2.645 | 0.108  | 0.946| 26.720  | 3.300| 4.091|
> | **RFWave** | 18M| 10 day (4×A100) | 1.280 | 3.020 | 0.078 | 0.957 | 18.126  | 2.988  | 4.169 |
> | **PeriodWave**   | 31M  | 1M steps, 4 day (4×A100)| 0.929  | 3.644   | 0.070  | 0.968 | 19.538  | 3.581   | 4.352   |
>
>
> **<Table R5-2: Band-wise Comparison of the reconstruction from EnCodec tokens. We utilize speech dataset from the test samples provided in RFWave.>**
>
> | Method | Params | Training | Mel-L (↓) | Mel-M (↓) | Mel-H (↓) | Mel-A (↓) |
> | --- | --- | --- | --- | --- | --- | --- |
> | **MBD** | 411M | 4×2day (4×V100) | 0.837 | 0.994 | 1.107 | 0.921 |
> | **RFWave** | 18M | 10 day (4×A100) | 0.603 | 0.674 | 0.775 | 0.651 |
> | **PeriodWave** | 31M | 1M steps, 4 day (4×A100) | 0.386 | 0.365 | 0.413 | 0.387 |
>
> **<Table R5-3: Objective evaluation results from EnCodec tokens. We utilize vocal dataset from the test samples provided in RFWave.>**
>
> | Method | Params | Training | M-STFT (↓) | PESQ (↑) | Period. (↓) | V/UV (↑) | Pitch (↓) | Mel-L (↓) | Mel-M (↓) | Mel-H (↓) | Mel-A (↓) |
> | --- | --- | --- | --- | --- | --- | --- | --- | --- | --- | --- | --- |
> | **MBD** | 411M | 4×2day (4×V100) | 1.554 | 2.894 | 0.096 | 0.957 | 22.252 | 0.891 | 1.024 | 1.095 | 0.957 |
> | **RFWave** | 18M | 10 day (4×A100) | 1.345 | 2.878 | 0.079 | 0.962 | 18.849 | 0.760 | 0.690 | 0.701 | 0.735 |
> | **PeriodWave** | 31M | 1M steps, 4 day (4×A100) | 0.921 | 3.681 | 0.067 | 0.967 | 17.531 | 0.448 | 0.391 | 0.415 | 0.431 |
>
> **<Table R5-4: Objective evaluation results from EnCodec tokens. We utilize sound effect dataset from the test samples provided in RFWave.>**
>
> | Method | Params | Training | M-STFT (↓) | PESQ (↑) | Mel-L (↓) | Mel-M (↓) | Mel-H (↓) | Mel-A (↓) |
> | --- | --- | --- | --- | --- | --- | --- | --- | --- |
> | **MBD** | 411M | 4×2day (4×V100) | 1.714 | 1.936 | 0.914 | 1.080 | 1.054 | 0.974 |
> | **RFWave** | 18M | 10 day (4×A100) | 1.592 | 2.253 | 0.701 | 0.640 | 0.705 | 0.690 |
> | **PeriodWave** | 31M | 1M steps, 4 day (4×A100) | 1.101 | 2.699 | 0.438 | 0.398 | 0.395 | 0.422 |
>
> Following RFWave, we utilize the same test set provided in RFWave as below:
>
> | **Category** | **Subdirectory** | **Samples** |
> | --- | --- | --- |
> | **Speech** | Expresso-test | 50 |
> |  | HiFiTTS-test | 50 |
> |  | LibriTTS-test | 50 |
> |  | Aishell3-test | 50 |
> |  | JVS | 50 |
> |  | CML-TTS-test | 50 |
> | **Vocal** | Musdb-test-vocal | 60 |
> |  | CSD | 20 |
> |  | Opencpop | 20 |
> |  | ChineseOpera-monophonic | 60 |
> |  | JVS-Music | 60 |
> |  | RAVDESS | 60 |
> |  | Ccmusic-demo-vocal | 20 |
> | **Sound Effect** | ESC-50 | 150 |
> |  | Musdb-test-accompaniment | 40 |
> |  | Musdb-test-mixture | 20 |
> |  | ChineseOpera-polyphonic | 20 |
> |  | OpenMIC-test | 40 |
> |  | Ccmusic-demo-music | 30 |

---

> ### Author Response · Authors · 2024-11-23
> **Thanks for your Reply (part2)**
>
> **[What makes their method better than the MBD?]**
>
> We have conducted extensive ablation studies to verify the effectiveness of our models, including the optimization of CFM objective and our multi-period structure for waveform signal at the submitted paper. Furthermore, we added the ablation study between CFM and Diffusion.
>
> In response to your question, the diffusion process in MBD is identical to the model described in our prior feedback. The only difference is the noise scheduling method with power schedule instead of cosine schedule. Furthermore, we utilize a much more simplified path $x_1-x_0$ during training. Additionally, CFM was well known that accelerate the training speed in terms of convergence speed, and performance with a simplified path. However, previous model architectures were inadequate for modeling high-resolution waveform signals. To address this limitation, we proposed a multi-period generator designed specifically to optimize high-resolution waveform modeling. By integrating CFM with a well-designed generator, we demonstrate that the overall performance of the model can be significantly improved.
>
> **[Regarding training speed]**
>
> We can discuss the fast training speed of our model in terms of the detailed GAN requirements, the effectiveness of CFM, and our efficient shallow structures.
>
> **Faster training speed in terms of the detailed GAN requirements**
>
> We can provide the additional requirements for GAN training as below:
>
> **<Table R6-1: The detailed GAN requirements>**
>
> |  | Generator | Discriminator | Number of Disc. | Discriminator Feed-forward | Total Disc. Feed-forward |
> | --- | --- | --- | --- | --- | --- |
> | PeriodWave | 29M | No Discriminator | 0 | 0 | 0 |
> | BigVGAN | 112M | MPD5, MRD3  | 8 |Real\*2, Fake\*2 (Disc, Generator, FM Loss) | 8*4 for 1 step |
> | BigVSAN | 112M | MPD5, MRD3 | 8 | Real\*2, Fake\*2 (Disc, Generator, FM Loss) | 8*4 for 1 step |
> | HiFi-GAN | 14M | MPD5, MSD3 | 8 | Real\*2, Fake\*2 (Disc, Generator, FM Loss) | 8*4 for 1 step |
>
> **Faster training speed in terms of CFM objective compared to GAN**
>
> As provided in our results, our model is trained with a single CFM Objective, which makes optimization significantly simpler compared to GAN-based approaches. GAN requires optimization of multiple loss terms, including Reconstruction Loss, GAN Loss, Feature Matching Loss, and Discriminator Training. This complexity increases the number of steps required to effectively optimize the generator.
>
> **Faster training speed in terms of Model Architecture**
>
> Our model achieves faster training speed due to its highly efficient architectural design. Specifically, we adopted a much smaller hidden size for processing high-resolution features and minimized the number of convolutional layers compared to HiFi-GAN and BigVGAN. Furthermore, we employed a downsampling strategy with strides of [4, 4, 4] in each block, enabling efficient parallel computation and lower memory consumption.
>
> This architectural optimization provides clear advantages: our model requires only 9GB of memory, compared to 62GB for BigVGAN and 26GB for HiFi-GAN under the same experimental setup. With fewer layers and reduced computational complexity, our model converges faster during training while maintaining competitive performance. Experimental results demonstrate that our architecture achieves an excellent balance between efficiency and effectiveness, leading to a notable boost in training speed compared to other structures.
>
> **<Table R6-2: The number of layer for generator>**
>
> |  | Generator | Number of Convolutional layer | Pre, post layer | up/down sampling layer |
> | --- | --- | --- | --- | --- |
> | PeriodWave | 29M | 36 layers | 2 layers | 6 (3 + 3) |
> | BigVGAN | 112M | 108 layers | 2 layers | 6 |
> | BigVSAN | 112M | 108 layers | 2 layers | 6 |
> | HiFi-GAN | 14M | 72 layers | 2 layers | 4 |

---

> > ### Author Response · Authors · 2024-11-23
> >
> > If there are any remaining uncertainties or points that require further clarification, we would be pleased to discuss them further to ensure complete understanding.
> >
> > Additionally, we believe this discussion underscores the robustness of our approach and hope it contributes positively to your overall evaluation.

---

> ### Author Response · Authors · 2024-11-23
>
> We would like to highlight the ablation study once again to address your concerns regarding why our structure have shown the powerful performance.
>
> In the submitted paper, we conducted ablation studies for multi-period structures. Introducing period-aware structure with multiple periods can significantly improve the performance compared to the model that utilizes a single period. Additionally, we included further analysis of different periods the reviewer nYdJ suggested ([1,1,1,1,1], [1,1,1,3,3], and [1,1,1,3,9]). This result also demonstrated that using different periods could enhance the generalization of waveform modeling.
>
> **[About the effectiveness of Multi-Period Structure]**
>
> **[Table R7]** Comparison of models trained with different periodic features.
>
> | Method | Period | M-STFT (↓) | PESQ (↑) | Periodicity (↓) | V/UV F1 (↑) | UTMOS (↑) |
> | --- | --- | --- | --- | --- | --- | --- |
> | Ground Truth | - | - | - | - | - | 3.8626 |
> | PeriodWave | [1,2,3,5,7] | 1.1737 | 4.072 | 0.0806 | 0.9627 | 3.5544 |
> |  | [1] | 1.2588 | 3.795 | 0.0885 | 0.9572 | 3.4215 |
> |  | [1,1,1,1,1] | 1.1337 | 3.964 | 0.0888 | 0.9597 | 3.4728 |
> |  | [1,1,1,3,3] | 1.1234 | 4.011 | 0.0818 | 0.9643 | 3.4879 |
> |  | [1,1,1,3,9] | 1.2736 | 4.061 | 0.0830 | 0.9644 | 3.5057 |
> |  | [1,2,4,6,8] | 1.1481 | 4.075 | 0.0782 | 0.9647 | 3.5468 |
> |  | [1,2,4,8,16] | 1.1463 | 4.124 | 0.0787 | 0.9639 | 3.5408 |
> |  | [1,2,3,5,7,11,13,17] | 1.1617 | 4.125 | 0.0792 | 0.9610 | 3.5384 |
>
> ---
>
> it is worth noting that the model with a single period is a similar with MBD model that use CFM and single-band structure. This comparison clearly highlights the effectiveness of our multi-period structures. Furthermore, we have added an ablation study for CFM and Diffusion demonstrating that optimizing the CFM can also enhance overall performance.
>
> Additionally, we have conducted ablation study for additional model structure, temperature for sampling, and FreeU for waveform generation at the original submission. All the proposed methods contribute to improvements in both performance and training speed in terms of convergence.

---

> ### Author Response · Authors · 2024-11-23
>
> **[Additional analysis on RFWave style waveform generation]**
>
> RFWave is another paper using CFM for waveform generation, submitted in the same conference (ICLR 2025). https://openreview.net/forum?id=gRmWtOnTLK
>
> This paper also demonstrate the effectiveness of CFM for waveform modeling, effectively incorporating CFM, iSTFT-based modeling, and multi-band structures to achieve high-quality and fast waveform generation. RFWave focuses on fast and efficient modeling on the complex spectrogram. We think that this approach is very cool idea for waveform modeling.
>
> However, we have also analyzed the performance of iSTFT-based CFM models in our preliminary studies last year. In our results, iSTFT-based waveform modeling significantly underperform compared to signal-level waveform modeling without multi-band modeling.
>
> **<Table R7-1: Ablation Study of model architecture on the dev subsets of LibriTTS. We trained each model for 1.0M steps with the same setting of PeriodWave.>**
>
> | **Method** | **Training Steps** | **M-STFT (↓)** | **PESQ (↑)** | **Periodicity (↓)** | **V/UV F1 (↑)** | **Pitch (↓)** | **UTMOS (↑)** |
> | --- | --- | --- | --- | --- | --- | --- | --- |
> | PeriodWave (16 steps) | 1.0M | 1.2129 | 4.224 | 0.0762 | 0.9652 | 17.496 | 3.6495 |
> | PeriodWave (16 steps, FreeU) | 1.0M| 1.0360 | 4.245 | 0.0765 | 0.9651 | 17.398 | 3.7307 |  |
> | Our iSTFT model (16 steps) | 1.0M | 1.1983 | 3.909 | 0.0849 | 0.9595 | 23.073 | 3.3123 |
> | Our iSTFT model  without U-ConvNeXt | 1.0M | 1.2324 | 3.861 | 0.0853 | 0.9600 | 19.343 | 3.3411 |
> | Our iSTFT model  with Middle Block Conditioning | 1.0M | 1.2307 | 3.895 | 0.0881 | 0.9590 | 25.080 | 3.2889 |
> | Our iSTFT model  without  Gate Modulation | 1.0M | 1.2080 | 3.818 | 0.0867 | 0.9571 | 26.774 | 3.2262 |
> | Our iSTFT model  without  Mel-Encoder | 1.0M | 1.3017 | 3.632 | 0.0992 | 0.9491 | 30.989 | 2.9847 |
> | Our iSTFT model  with iSTFT Head of Vocos | [Unstable] | - | - | - | - | - | - |
> | Only ConvNeXt-v2 | 1.0M | 1.5512 | 2.750 | 0.1219 | 0.9308 | 24.784 | 2.7893 |
>
> First, we would like to share our experiments for iSTFT-based waveform generation. We significantly improve the performance by adding several techiques including Unet-style ConvNeXt, AdaLN Style Gate modulation in ConvNeXt block, Mel-Encoder (proposed in PeriodWave), and different style of iSTFT head.
>
> While iSTFT models have faster inference speed, they still significantly underperform compared to PeriodWave
>
> Furthermore, we would like to share the scalability of iSTFT-based model as below:
>
> **<Table R7-2: Comparison on the effectiveness of depth and width of ConvNext-v2-based waveform generation. We trained each model for 1.0M steps with the same setting of PeriodWave.>**
>
> | **Depth (Blocks)** | **Width (Hidden Dim.)** | **Params.** | **M-STFT (↓)** | **PESQ (↑)** | **Periodicity (↓)** | **V/UV F1 (↑)** | **Pitch (↓)** | **UTMOS (↑)** |
> | --- | --- | --- | --- | --- | --- | --- | --- | --- |
> | 16 | 1024 | 219.35M | 1.1393 | 4.114 | 0.0769 | 0.9651 | 20.866 | 3.6192 |
> | 16 | 512 | 55.67M | 1.1920 | 3.987 | 0.0841 | 0.9589 | 23.675 | 3.4490 |
> | 12 | 768 | 102.71M | 1.1335 | 4.058 | 0.0790 | 0.9627 | 20.863 | 3.5441 |
> | 8 | 1024 | 143.68M | 1.1925 | 4.015 | 0.0840 | 0.9610 | 19.191 | 3.5033 |
> | 8 | 512 | 35.56M | 1.1983 | 3.909 | 0.0849 | 0.9595 | 23.073 | 3.3123 |
> | 6 | 384 | 18.27M | 1.2879 | 3.442 | 0.1094 | 0.9456 | 28.958 | 2.7703 |
> | 6 | 256 | 8.00M | 1.6696 | 1.356 | 0.1863 | 0.8822 | 74.895 | 1.3800 |
>
> We found that dimension size of ConvNeXt is crucial to ensuring the performance of iSTFT-based model, requiring a size greater than 384 to effectively generate waveform signal.
> While the model with 219.35M parameters performed better than other configurations, iSTFT-based model still underperform compared to PeriodWave.
>
> In this regard, we introduce multi-period structure for the waveform-level modeling. However, we also acknowledge that RFWave is an excellent paper that explores different approaches to optimize iSTFT-based waveform modeling. We believe that both CFM-based waveform generation models definitely contribute to the speech and audio research by providing innovative approaches to waveform modeling.
>
> We are eager to address your concerns. If you have any additional questions, please let us know, and we will be happy to provide further clarification.
>
> Best Regards
>
> Authors

---

> ### Author Response · Authors · 2024-11-25
> **We would like to follow up on our rebuttal**
>
> Dear **Reviewer TzrS,**
>
> We sincerely thank you once again for your valuable time and effort in reviewing our manuscripts.
>
> We deeply appreciate your insightful comments and we have worked to address your concern through additional experiments and clarifications. We outline them as below:
>
> - **[Fair Comparison with MBD and RFWave]** As you requested, we conducted additional experiments by comparing our models with MBD and RFWave. Using the same training dataset and the same universal test dataset provided in RFWave, we ensured fairness and consistency throughout the evaluations.
> - **[SOTA Performance in all evaluations]** We have demonstrated the superior performance of our model through extensive experiments. Compared to MBD (NeurIPS, 2023), BigVGAN (ICLR, 2023), and Vocos (ICLR, 2024), our model achieves state-of-the-art results. Furthermore, our model outperformed RFWave (Submitted in ICLR 2025), which received an average score of 6 (Submitted in ICLR2025). We hope this provided additional context when assessing the potential of our work.
> - **[Comparison with CFM and Diffusion]** As you requested, we demonstrated the effectiveness of CFM for waveform modeling within the same experimental setup.
> - **[Fast Convergence Speed]** As you questioned, we have provided a detailed analysis explaining the factors contributing to our model’s faster training speed and faster convergence compared to others.
>
> To address your concerns thoroughly within the limited discussion time, we undertook significant additional experiments, incurring **substantial costs to access high-performance GPUs**. This reflects our commitment to ensuring the robustness and clarity of our submission.
>
> We hope these comments and improvements can be taken into account when deciding on the final score for paper. We remain open to further discussion and are happy to provide additional clarifications if needed.
>
> Thank you once again for your constructive feedback and the opportunity to improve our work.
>
> sincerely,
>
> The Authors.

---

> > ### Comment · Reviewer_TzrS · 2024-11-25
> > **Official Comment by Reviewer TzrS**
> >
> > I would like to thank the authors for providing additional details, clarifications and empirical results. I see now that the comparison to MBD and RFWave seems to be fair.
> > I highly encourage the authors to include such results in the main paper. In terms of modeling architecture, I suggest the authors include such details and results in the paper as well, to support your run time efficiency with empirical evidence. I've increased my score accordingly.

---

> > > ### Author Response · Authors · 2024-11-26
> > >
> > > Thank you very much for your feedback and for significantly improving our score. We deeply appreciate your contribution, as your suggestions have greatly strengthened the quality of our manuscripts.
> > >
> > > We will incorporate the details and results of the new experiments into the main manuscripts as you recommended
> > >
> > > Sincerely,
> > >
> > > The Authors

---

### Author Response · Authors · 2024-11-24
**Global Response by Authors**

Dear Reviewers,

We thank the reviewers for taking the time to carefully read our manuscript, as well as for their valuable comments and constructive suggestions.

We greatly appreciate the reviewers’ recognition of **PeriodWave** as a novel universal waveform generator that achieves superior performance compared to the baselines (**TzrS, HbGY, cFmr, nYdJ, C9NL**). Additionally, we are encouraged that the reviewers acknowledged this as the first application of **Flow Matching** to waveform generation (**HbGY, nYdJ**).

We are pleased that the reviewers found our paper to include extensive experimental work, demonstrating the robustness of the proposed model in various scenarios, such as using Mel-spectrogram and discrete representation extracted from neural codecs, the streaming decoding strategy (**C9NL**), achieving a good balance between quality and synthesis speed compared to conventional diffusion-based methods (**cFmr**), and the novel FreeU analysis for waveform generation (**HbGY, C9NL**).

We have updated our manuscript as follows:

- **Pg. 28-29**: We have added **additional experiments for waveform generation from EnCodec tokens** to demonstrate superior performance compared to SOTA model, MBD and concurrent work RFWave, which was submitted to ICLR2025. We trained the model using the same training dataset summarized in MBD and RFWave to ensure a fair comparison. Furthermore, we evaluated each model using the universal evaluation dataset provided by RFWave, specifically, conducting evaluations for **speech**, **vocal**, and **sound effects**, respectively.
- **Pg. 28**: We have Included **band-wise comparison metrics** as suggested by reviewers (**cFmr**, **nYdJ**).
- **Pg. 30**: We have added **detailed spectrograms** for each model, as suggested by reviewer **cFmr**.
- **Pg. 31**: We have conducted additional ablation studies for **diffusion** and **CFM**, as suggested by reviewers **(TzrS, nYdJ)**.
- **Pg. 31**: We have added additional **ablation studies for different periods**: ([1,1,1,1,1], [1,1,1,3,3],[1,1,1,3,9]), as suggested by reviewer **nYdJ**.
- **Pg. 32**: We have added results of **PeriodWave-MB with FreeU** and additional analysis of FreeU as suggested by reviewer **cFmr**
- **Demo Page**: We have added the samples of new experiments at the demo page.

We sincerely appreciate the reviewers’ valuable feedback and are eager to address any concerns you may have. If you have further questions or require additional clarification, please let us know.

Best regards,

The Authors

---

### Meta-Review · Area_Chair_Y8YZ · 2024-12-22

**Metareview:**

> This paper studies the task of speech and audio waveform generation using the flow-matching modeling paradigm. The authors proposed a multi-period approach to generation artifacts in high-frequencies, conditioned on either mel-spectrogram and discrete representation extracted from neural codecs. The authors compared the proposed method to several GAN-based approaches and two diffusion-based methods. The authors empirically demonstrated the proposed method reaches superior performance to the evaluated baselines.

The reviewers are unanimous about acceptance. This was a borderline paper that improved significantly through the discussion between reviewers and authors during the rebuttal period.

**Additional Comments On Reviewer Discussion:**

This was a borderline paper that improved significantly through the discussion between reviewers and authors during the rebuttal period.

---

### Decision · Program_Chairs · 2025-01-22

Accept (Poster)